# Distinct evolution of SARS-CoV-2 Omicron XBB and BA.2.86/JN.1 lineages combining increased fitness and antibody evasion

Delphine Planas [1,2,13] ✉, Isabelle Staropoli[1], Vincent Michel [3], Frederic Lemoine [4,5], Flora Donati [4,6], Matthieu Prot[4], Francoise Porrot [1], Florence Guivel-Benhassine[1], Banujaa Jeyarajah[6], Angela Brisebarre[6], Océane Dehan[6], Léa Avon[6], William Henry Bolland[1], Mathieu Hubert [1], Julian Buchrieser [1], Thibault Vanhoucke[1], Pierre Rosenbaum [7], David Veyer[8,9], Hélène Péré[8,9], Bruno Lina [10,11], Sophie Trouillet-Assant[10,11], Laurent Hocqueloux[12], Thierry Prazuck[12], Etienne Simon-Loriere [4,6,13] ✉ & Olivier Schwartz [1,2,13] ✉

The unceasing circulation of SARS-CoV-2 leads to the continuous emergence of novel viral sublineages. Here, we isolate and characterize XBB.1, XBB.1.5, XBB.1.9.1, XBB.1.16.1, EG.5.1.1, EG.5.1.3, XBF, BA.2.86.1 and JN.1 variants, representing >80% of circulating variants in January 2024. The XBB subvariants carry few but recurrent mutations in the spike, whereas BA.2.86.1 and JN.1 harbor >30 additional changes. These variants replicate in IGROV-1 but no longer in Vero E6 and are not markedly fusogenic. They potently infect nasal epithelial cells, with EG.5.1.3 exhibiting the highest fitness. Antivirals remain active. Neutralizing antibody (NAb) responses from vaccinees and BA.1/BA.2-infected individuals are markedly lower compared to BA.1, without major differences between variants. An XBB breakthrough infection enhances NAb responses against both XBB and BA.2.86 variants. JN.1 displays lower affinity to ACE2 and higher immune evasion properties compared to BA.2.86.1. Thus, while distinct, the evolutionary trajectory of these variants combines increased fitness and antibody evasion.

Succeeding sub-lineages of Omicron have spread since the appearance of BA.1 in November 2021[1,2]. More than 90% of the human population have been probably infected by one Omicron subvariant, in the absence of an efficient and long-lasting protection against novel viral acquisitions conferred by previous infections or vaccinations[3–6], allowing the virus to further evolve and diversify. Identified in September 2022, the XBB lineage originated from a recombination of two BA.2-derived variants (BJ.1 and BM.1.1.1) and progressively replaced most of previous Omicron strains. Members of this lineage are characterized by enhanced transmissibility rates and immune evasion properties[7,8]. These variants are responsible for small waves of

contaminations in many countries. Their geographical distribution is somewhat heterogeneous. The variants are closely related and carry an additional and limited set of mutations in the spike corresponding to a stepwise accumulation of changes. Convergent evolution may have been associated with this process. For instance, many lineages independently acquired mutations in the Receptor Binding Domain (RBD) of the spike, including F486P or F456L, that are known escape mutations for neutralizing antibodies[9,10]. Other recombinants increased in frequency in regions of the world but did not spread extensively, for example XBF (a recombinant of BA.5.2.3 and BA.2.75.3 lineages) or XBC (a recombinant of BA.2 and Delta lineages), both noted in Australia or

New Zealand and carrying the F486P substitution. This convergent evolution is likely due to a similar selective pressure exerted by imprinted or hybrid immunity triggered by Omicron infection and/or vaccination[11–14].

In August 2023, a lineage named BA.2.86 corresponding to an important evolutionary jump has been detected in multiple countries, prompting its classification as a Variant of Interest by the World Health Organization[15]. The effective reproduction number of BA.2.86 is estimated to be higher or similar than those of XBB.1.5 and EG.5.1[16]. A high attack rate of BA.2.86 (above 85%) occurred in a large care home outbreak, confirming its high transmissibility[17]. There is so far no clinical evidence of increased pathogenicity of BA.2.86. In hamsters, BA.2.86 displays an attenuated phenotype[18,19]. The impact of this unprecedented combination of mutations on antibody evasion has started to be deciphered. A few recent articles and preprints reported that NAb responses to BA.2.86 are lower than BA.2, but comparable or slightly higher than to other simultaneously circulating XBB-derived variants[16,19–30]. Most of these studies were performed with lentiviral or VSV pseudotypes. Isolation of a BA.2.86 virus confirmed the antibody-escape properties of this strain[20,24]. BA.2.86 spike displays stronger affinity to ACE2 than other variants[27,28,30], but the consequences on viral replication and tropism remain poorly understood.

The BA.2.86 lineage has then rapidly started to evolve, with the emergence of the JN.1 sub-lineage in September 2023. In December 2023, JN.1 sharply increased in frequency in Europe and USA. It became predominant worldwide and was the main variant responsible for the epidemic surge in December 2023-January 2024. Based on epidemiological modeling, it has been estimated that JN.1 displays a 2.3-fold growth advantage relative to EG.5.1.1[31]. A preprint using JN.1 pseudotypes reported enhanced immune evasion properties, particularly to class 1 neutralizing antibodies, associated with a decreased affinity to ACE2, relative to BA.2.86[32].

Here, we isolated and characterized 9 viral strains that were circulating late 2023-early 2024. We performed a side-by-side comparison of their replication in cell lines and relevant human primary nasal epithelial cells, their binding to ACE2 and fusogenicity. We examined their sensitivity to previously approved mAbs and antiviral drugs, to sera from recipients of various vaccine regimens, and to individuals who experienced breakthrough infections during XBB circulation.

## Results

### Mutations in XBB-derived and BA2.86 spike sequences

XBB variants replaced previously circulating Omicron variants in early 2023 and have been continuously evolving. In October 2023, the main XBB sub-variants, representing about 80% of reported viral sequences, were XBB.1.5, XBB.1.9.1, XBB.1.16, XBB.2.3 and the EG.5.1 sublineage (Fig. 1a). EG.5.1 is a descendant of XBB.1.9.2 that was designated in May 2023 and has since then been on the rise[33]. The set of the spike mutations is depicted Fig. 1b and the resulting phylogenetic tree is displayed Fig. S1a. In addition to the R346T and N460K substitutions, these most frequent XBB lineages independently acquired the S486P substitution. Other substitutions were noted at this position in previous variants, such as 486 V in BA.5 (Fig. S1b). These variants have spread worldwide, with local variations in frequency. The subsequent sequential acquisition of the F456L substitution has been repeatedly noted in 486 P carrying XBB lineages, suggesting an epidemiological advantage for their combination in this genomic background. More than 65% of genomes shared on GISAID in October 2023 carry both F486L and F456L substitutions.

In contrast to the steady accumulation of substitutions observed in XBB sublineages, the evolutionary processes that led to the emergence of BA.2.86 have not been captured. BA.2.86 spike carries a novel and distinct constellation of changes. This leads to a novel branch in the SARS-CoV-2 spike phylogenetic tree (Fig. S1a). Compared with its BA.2 ancestor, the spike contains 34 changes, a number comparable to the difference between the initial Wuhan virus and BA.1, including novel insertions, deletions, and substitutions spanning the whole protein (Fig. 1b). Some mutations, such as G446S, N460K, F486P, and R493Q have been reported in other variants[7,8,34] (Fig. S1b), but others are less frequent and poorly characterized. Furthermore, BA.2.86 has been quickly diversifying as it spreads. BA2.86.1 is characterized by two novel mutations in ORF1. JN.1 carries one additional amino acid substitution (F455S) in the spike (Fig. S1b) along with the ORF1a:R3821K and ORF7b:F19L changes (Fig. S2).

### Isolation of 9 variants circulating in late 2023-early 2024

We isolated XBB.1, XBB.1.5, XBB.1.9.1, XBB.1.16.1, XBF, EG.5.1.1, EG.5.1.3, BA.2.86.1 and JN.1 variants from nasopharyngeal swabs received at the French National Reference Center of Respiratory viruses (Paris, Lyon) and Hôpital Européen Georges Pompidou (Paris, France), using either a Vero E6 TMPRSS2+ clone (thereafter termed Vero E6 TMP-2 cells) or IGROV-1 cells. We reported that the IGROV-1 human ovarian cell line is highly sensitive to SARS-CoV-2 and produces high levels of infectious virus[35]. After isolation, the 9 viruses were thus amplified by one passage on IGROV-1 cells. The sequences of the variants after amplification were identical to those obtained from the primary samples (see GISAID accession numbers in Table S3), indicating that no mutations occurred during this short culture period. The spike mutations in these 9 variants, compared to BA.2, are depicted in Fig. 1b. We also compared the spike mutations of a larger panel of variants to the Wuhan ancestral strain in Fig. S1b.

### Replication of XBB-derived, BA2.86.1 and JN.1 variants in Vero E6 derivatives and IGROV-1 cells

We characterized the fitness of the 9 variants by assessing their replication in different cells and adding as controls D614G, BA.1, BA.5 or BQ.1.1. Since JN.1 was isolated later than the other variants, it was included in most but not all experiments. We first chose Vero E6 and IGROV-1 cells, that both naturally express ACE2, but not TMPRSS2, the protease that primes SARS-CoV-2 for fusion, as verified by flow cytometry (Fig. S3). Vero E6 cells efficiently replicate pre-Omicron strains but are less sensitive to previous Omicron variants[35]. We thus compared the permissibility of Vero E6 and IGROV-1 cells to all variants (Fig. S4a). Viral stocks were serially diluted and incubated with the two target cells. After 48 h, cells were stained with an anti-SARS-CoV-2 N monoclonal antibody. Foci of infected cells were automatically scored (Fig. S4b). The ancestral D614G strain was similarly infectious in the two cell lines (Fig. S4). In contrast, none of the 12 Omicron variants efficiently infected Vero E6 whereas they were highly infectious in IGROV-1. We then asked whether the poor infectivity of the variants in Vero E6 was due to the lack of TMPRSS2. We thus selected two Vero E6 subclones engineered to express TMPRSS2. The first clone (termed Vero E6 TMP-1) was generated in our laboratory[35] and expresses high levels of TMPRSS2 and rather low levels of ACE2, probably because this receptor can be cleaved by the protease[36] (Fig. S3). The Vero E6 TMP-2 was previously described[37]. It expresses low surface levels of TMPRSS2 and ACE2 levels comparable to those in Vero E6 cells (Fig. S3). We thus compared the kinetics of viral replication in IGROV-1, Vero E6, Vero E6-TMPRSS2 clones 1 and 2. We used Delta and BQ.1.1 as controls (a BA.5-derived Omicron variant), and compared them to XBB.1, XBB.1.5, EG.5.1.3 and BA.2.86.1 (Fig. 2a). All viruses efficiently replicated in IGROV-1 cells, with a peak of infected cells detected at day 2 post infection (p.i.). The Omicron variants did not potently infect Vero E6 during the 4 day survey period. The two Vero E6-TMPRSS2 clones behaved differently regarding their sensitivity to variants. Vero E6 TMP-1 did not support strong replication of Omicron variants, despite allowing growth of Delta. In contrast, Vero E6 TMP-2 efficiently replicated the novel variants (Fig. 2a). The differences between Vero E6 TMP-1 and TMP-2 might be due to clonal effects. Alternatively, high levels of TMPRSS2, and reduced surface expression of ACE2 in Vero E6

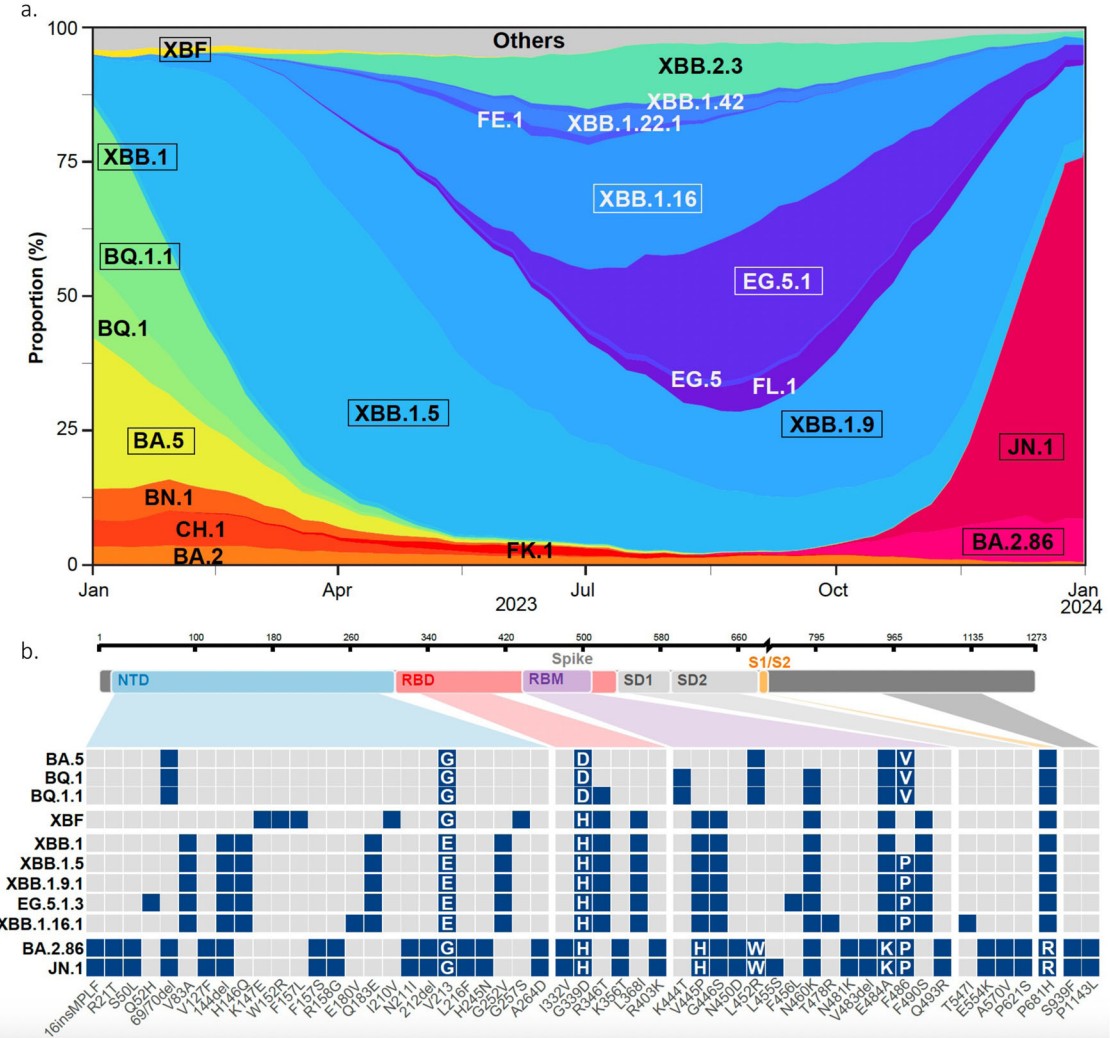

**Fig. 1 | SARS-CoV2 evolution in 2023 and Spike mutation patterns of the main lineages. a** Evolution of the prevalence of main SARS-CoV-2 lineages from January to December 31, 2023. The pattern of emergence and replacement of several lineages, such as XBB.1.5, then XBB.1.9, XBB.1.16 or EG.5.1 and the emergence of BA.2.86 and JN.1 is shown. The variants with a frame are analyzed in this study. **b** Changes specific to lineages studied here in comparison to BA.2 are displayed as colored squares. The spike domain organization is displayed on the top, with N-terminal domain (NTD), Receptor Binding Domain (RBD), Receptor Binding Motif (RBM), Single domains SD1 and SD2, S1/S2 cleavage site, and S2 domains. BA.2.86 and its descendant JN.1 show many new mutations compared to other lineages. A complete comparison of spike mutations compared to the reference Wuhan_Hu-1 is presented in Fig. S1b.

TMP-1 might be detrimental for XBB-derived and BA.2.86.1 variants. Regardless of the underlying mechanisms, our results indicate that the recent XBB-derived and BA.2.86 variants display a different tropism than pre-Omicron viruses for some cell lines. Low expression of TMPRSS2 in Vero E6 cells is associated with permissibility of the cells to XBB-derived and BA.2.86.1 infection.

We further explored the mechanisms underlying the high sensitivity of IGROV-1 to SARS-CoV-2 replication. We examined viral entry pathways in these cells and performed infections in presence of either Camostat, a TMPRSS2 inhibitor, SB412515, a cathepsin L inhibitor, or E-64d, a pan-cysteine protease inhibitor acting mainly on endocytic proteases[38,39]. The drugs were added 2 h before infection and maintained for 24 h, before scoring infected cells (Fig. S5). We first selected a few pre-Omicron and recent Omicron variants, (D614G, Delta, BA.1, XBB.1 and XBB.1.16.1) and tested the effect of the drugs in IGROV-1, Vero E6 and Vero TMP-1 cells. Camostat (100 μM) did not inhibit viral replication in IGROV-1 and Vero E6 cells, in line with the absence of detection of TMPRSS2 in these cells by flow cytometry. Camostat inhibited viral replication by 50–80% in Vero E6 TMP-1 cells, confirming that TMPRSS2 facilitates viral entry when present in target cells.

There was no significant difference in the sensitivity of variants to Camostat in Vero E6 TMP-1 cells. SB412515 and E-64d (both at 10 μM) strongly inhibited viral infections in IGROV-1 cells (>90% inhibition for all variants) but were less efficient in Vero E6 cells (Fig. S5). This suggests that endocytic viral entry is particularly active in IGROV-1. With both SB412515 and E-64d, similar ED50 were obtained for D614G, XBB.1.5, EG.5.1.3 and BA.2.86.1 variants, which may indicate that all variants use similar entry pathways in IGROV-1 cells (Fig. 2b). In contrast to their strong antiviral effect in IGROV-1 and lower activity in Vero E6 cells, SB412515 and E-64d were almost inactive in Vero E6 TMP-1 cells, confirming that when TMPRSS2 is present, viral entry and fusion preferentially occurs at the cell surface[38,40]. We then assessed the sensitivity of Delta, BA.2.86.1 and JN.1 to Camostat (100 μM), SB412515, and E-64d (10 μM) in IGROV-1 and Vero E6 TMP-2 cells. SB412515 and E-64d effectively inhibited the replication of the three variants in IGROV-1 cells. Camostat inhibited the replication of Delta, BA.2.86.1 and JN.1 in Vero E6 TMP-2 cells, but not in IGROV-1 cells, confirming the results observed with the other variants (Fig. S6).

Therefore, IGROV-1 are highly sensitive to all SARS-CoV-2 strains, likely because of a strong endocytic pathway facilitating viral entry.

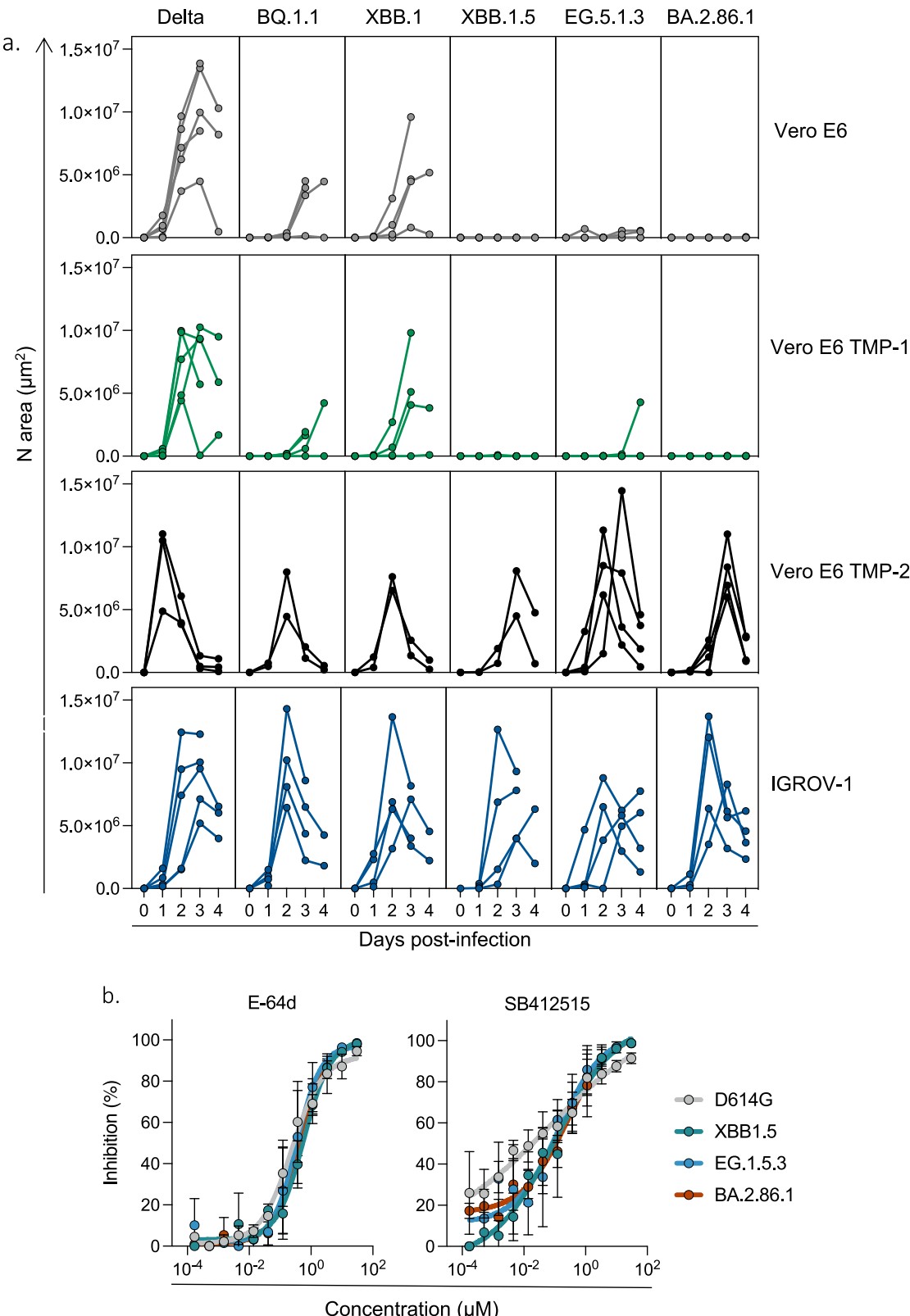

**Fig. 2 | Replication kinetics of SARS-CoV-2 variants in Vero E6, Vero E6 TMP-1 and -2 and IGROV-1 cells. a** Cells were infected with the indicated variants, at $3 \times 10^{-2}$ infectious units per cell. Cells were stained with a pan-coronavirus anti-N antibody at days 1–4 pi. The N-positive areas were plotted on the graph. Each curve represents an independent experiment. **b** Comparison of the effect of E-64d and SB412515 against different variants. IGROV-1 cells were pre-incubated 2 h with serial dilutions of E-64d or SB412515 ($30–1.7 \times 10^{-4}$ μM) and infected with D614G, XBB.1.5, EG.5.1.3, or BA.2.86.1. The percentage of inhibition is represented. Data are mean ± s.d. of 3 independent experiments. Source data are provided as a Source Data file.

## Fusogenicity and ACE2 binding of XBB-derived and BA2.86.1 variants

We next investigated the fusogenicity of the spikes and their ability to form syncytia independently of viral replication. We used a GFP-Split based model in which fused cells become GFP+ (Fig. 3a)[41,42]. We transfected the spikes into 293 T cells expressing the GFP-11 subunit and co-cultivated them with IGROV-1 cells expressing the GFP1-10 subunit (Fig. 3a). Spike expression on transfected HEK293T cells was similar across variants, as measured by staining with a pan-coronavirus anti-S2 mAb (Fig. S7). As previously reported by us and others, the ancestral D614G and Delta strains were more fusogenic than the early Omicron BA.1 and BA.4/5 variants[35,39,43,44]. BQ.1.1 and XBB.1.5/XBB.1.9.1 spikes partially regained fusogenicity, whereas XBB.1.16.1 and the more recent EG.5.1.1 and BA.2.86.1 spikes displayed a lower fusogenic potential than BA.4/5 spike (Fig. 3b, c). This profile of fusogenicity was similarly observed when Vero E6 cells were used as targets (Fig. 3d). With Vero E6 TMP-1 as target cells, the number of syncytia was increased by about 2-fold for all variants, when compared to Vero E6 cells (Fig. 3d), indicating that TMPRSS2 enhances their fusogenic activity. Thus, the recent XBB-derived and BA.2.86.1 variant spikes do not display high fusion properties, when compared to Delta or to their Omicron predecessors, at least in the cell lines tested.

We next examined the affinity of the different variant spikes to ACE2. To this aim, we infected IGROV-1 cells with most of the variants for 24 h. Cells were stained with an anti-N antibody to visualize productively infected cells and were exposed to serial dilutions of soluble biotinylated ACE2. Binding was measured by flow cytometry (Fig. S8a). ACE2 titration binding curves were generated, and EC50 (the amount of ACE2 needed for 50% binding) was calculated (Fig. S8b). The spikes of XBB.1, XBB.1.16.1 and EG.5.1.3 had comparable affinities to ACE2 than Delta, whereas BA.2.86.1 bound more potently to the receptor (Fig. 3e). Similar results were observed in 293 T cells transiently expressing the different spikes (Fig. 3f), confirming recent reports obtained with recombinant proteins[28,27].

## Replication of XBB-derived and BA2.86.1 variants in primary nasal epithelial cells

We used primary nasal epithelial cells (hNEC) grown over a porous membrane and differentiated at the air–liquid interface for 4 weeks (from MucilAirB™), to compare the different variants in a relevant model of SARS-CoV-2 infection[45,46,44]. The cells were infected with each variant at a similar low viral inoculum (100 μl of viral stocks containing $2 \times 10^3$ infectious units/ml). As reported[44,45,47,43], BA.1 replicated faster than Delta and D614G, as quantified by viral RNA (vRNA) release monitored every day up to 4 days pi (Fig. 4a, b). Compared to Delta, BA.1 displayed up to 60-fold increase in vRNA levels measured at 24 h pi. The XBB-derived variants exhibited a replication advantage compared to BA.1, with EG.5.1.3 displaying a 16-fold increase in vRNA release at 24 h. BA.2.86.1 replication kinetics resembled those of BA.1 and were not higher than other XBB-derived variants (Fig. 4a, b). Infectious virus release was monitored at 48 h p.i. (Fig. 4c) and correlated with vRNA levels. We assessed by immunofluorescence the cytopathic effect induced by the variants. Infected hNEC were stained at day 4 p.i. with anti-SARS-CoV-2 N antibodies, phalloidin (to visualize F-actin), anti-alpha tubulin antibodies (to visualize cilia) and anti-cleaved caspase 3 antibodies (to visualize apoptotic dying cells). When compared to Delta or BA.1, EG.5.1.3 and BA.2.86.1 displayed elevated markers of cytopathy, including disappearance of the ciliated structure and activation of caspase 3 (Fig. 4d; Fig. S9).

Therefore, Omicron variants, particularly BQ.1.1 and XBB-derived isolates, exhibit higher infectivity in hNECs compared to D614G and Delta. Among them, EG.5.1.3 demonstrates the highest fitness. Additionally, both EG.5.1.3 and BA.2.86.1 variants exhibit significant cytopathic effects in these cells (Fig. S9).

## Sensitivity of XBB-derived, BA2.86.1 and JN.1 variants to antiviral antibodies and small molecules

Several anti-spike monoclonal antibodies (mAbs) have been used as pre- or post-exposure therapy in individuals at risk for severe disease[10]. However, the first Omicron variants BA.1, BA.2 and BA.5 escaped neutralization from most of the mAbs, leading to changes in treatment guidelines[48]. For instance, as of mid-2022 and later, Ronapreve (Imdevimab + Casirivimab) or Evusheld (Cilgavimab + Tixagevimab) cocktails and Sotrovimab were no longer approved[48]. However, Sotrovimab retains some neutralizing and non-neutralizing Fc-mediated functions against BQ.1.1 and XBB.1.5[49]. We assessed with the S-Fuse assay the sensitivity of D614G, XBB.1.16.1, EG.5.1.3, BA.2.86.1 and JN.1 to Ronapreve, Evusheld or Sotrovimab. We included the ancestral D614G strain as a control, which was efficiently neutralized by the mAbs (Fig. 5). Evusheld and Ronapreve combinations were inactive against the recent variants. Sotrovimab remained weakly functional against XBB.1.16.1 and EG.5.1.3 but lost antiviral activity against BA.2.86.1 and JN.1. We examined whether the neutralization profile of Sotrovimab correlated with the ability of the mAb to bind the different spikes. We measured by flow cytometry the binding of Sotrovimab to IGROV-1 cells infected with the corresponding variants (Fig. S10a, b). Cells were stained with the pan coronavirus S2 antibody mAb10 as a control (Fig. S10c). The levels of binding with the different antibodies were quantified (Fig. S10d). Sotrovimab bound to all XBB-derived variants but not to BA.2.86.1. This is likely due to the presence of the K356T mutation in BA.2.86.1 RBD, that has been identified as conferring resistance to Sotrovimab[50,51].

We then assessed the efficacy of the currently approved anti-SARS-CoV−2 drugs Nirmatrelvir (present in Paxlovid), Remdesivir and Molnupiravir against D614G, XBB.1.5.1, EG.5.1.3 and BA.2.86.1. IGROV-1 cells were exposed to serial dilutions of the compounds, exposed to the variants, and infection was revealed after 24 h (Fig. 5b). The antiviral molecules remained active against the tested variants, with no significant differences in their EC50 (10 nM, 3.4 μM and 0.04 nM for Nirmatrelvir, Molnupiravir and Remdesivir, respectively).

## Cohort design

We collected 75 sera from two different cohorts representing a total of 41 vaccinated and/or infected individuals. The characteristics of the participants (dates of vaccination, breakthrough infection and sampling) are indicated in Table S1.

The first cohort includes two groups of individuals. The first group is composed of 21 health-care workers, in Orleans, France (Table S1a). The participants received two doses of Pfizer BNT162b2 vaccine and one or two booster doses with the same monovalent vaccine. 13 out of 21 individuals experienced a pauci-symptomatic breakthrough infection after the third injection. We did not generally identify the Omicron subvariant responsible for the breakthrough infections. 12 individuals were infected between December 2021 and mid-June 2022, a period when BA.1 and BA.2 were successively dominant in France[52]. One individual was infected in August 2022 and was likely positive for BA.5. All individuals received a Bivalent Wuhan/BA.5 Pfizer boost between December 2022 and February 2023. The second group includes 12 vaccinated health-care workers, in Orleans, France, that experienced a breakthrough Omicron infection in August/September 2023, when XBB-derived variants were predominant (Table S1c).

The second cohort includes 8 health-care workers, in Lyon, France, that were longitudinally sampled at day 0, 1 month and 6 months after their Bivalent Wuhan/BA.5 Pfizer boost (Table S1b).

## Sensitivity of XBB-derived and BA2.86.1 variants to sera from vaccinees

We assessed the sensitivity of the panel of XBB-derived and BA2.86.1 variants to serum samples from the two cohorts. We first asked whether antibodies elicited by three doses of the original Pfizer vaccine

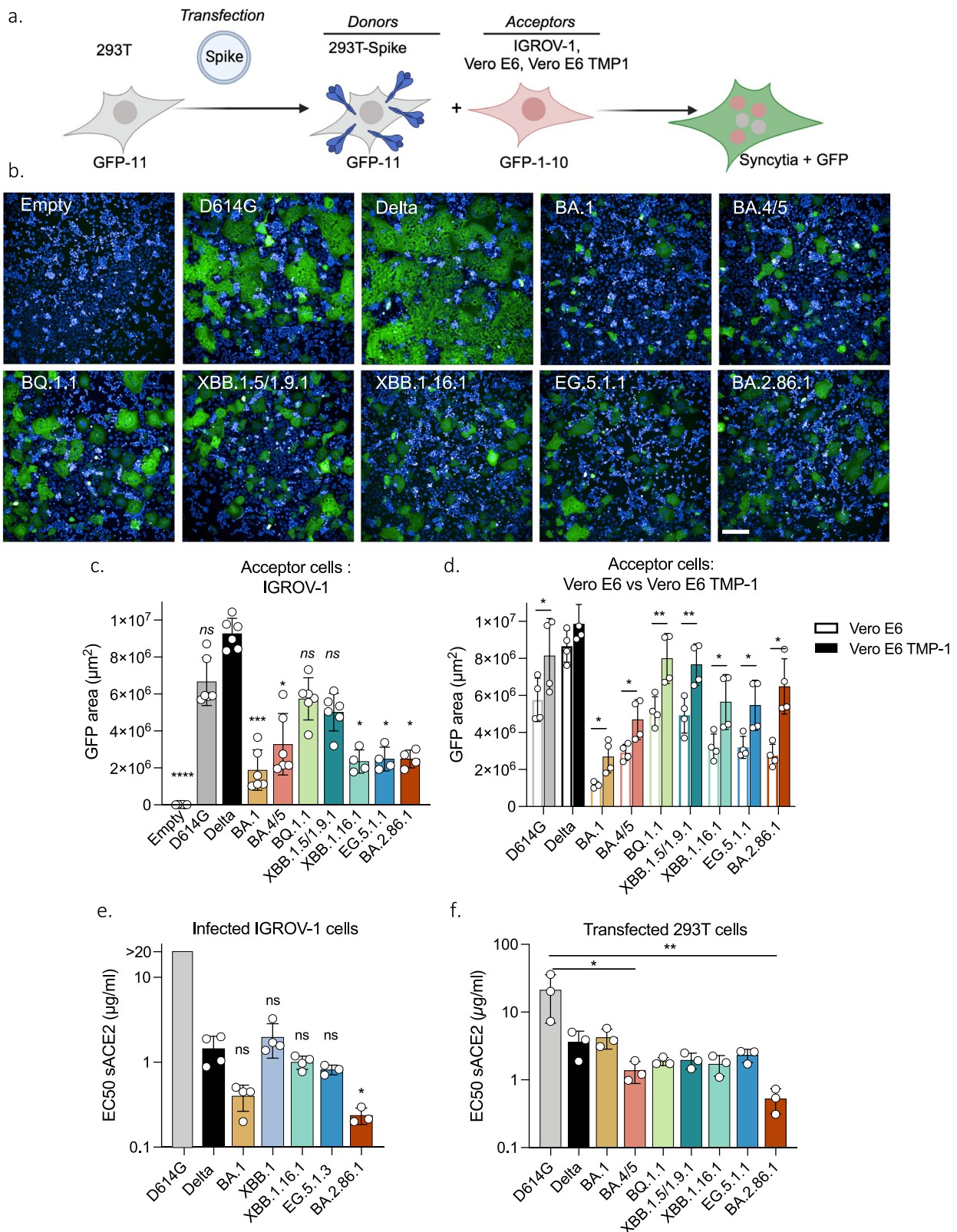

neutralized the novel subvariants. Twenty-one individuals were analyzed 12 months post third dose (Tables S1a, b, S2a). Among them, 12 individuals experienced a BA.1/BA.2 breakthrough infection. We measured the potency of their sera against BQ.1.1, XBB.1, XBB.1.5; XBB.1.9.1, XBB.1.16.1, XBF, EG.5.1.3 and BA.2.86.1. We used as reference the D614G ancestral strain, as well as BA.1 and BA.5 (Fig. 6a). We calculated the ED50 (Effective Dose 50%) for each combination of serum and virus.

The ED50 were high for D614G (ED50 of $6.7 \times 10^3$) and were decreased by 5-fold for BA.1 and BA.5, confirming the escape properties of these previous sublineages[35]. With the XBB-derived strains, the ED50 were low, ranging from $2 \times 10^2$ to $8 \times 10^1$. XBB.1.16.1 and EG.5.1.3 neutralization titers were the lowest (22 and 30-fold lower than BA.1, respectively). BA.2.86.1 neutralization was about 2-fold more sensitive to neutralization than EG.5.1.3, confirming recent results[16,20–29].

**Fig. 3 | Fusogenicity and binding to ACE2 of the variant spike proteins.**
**a** Schematic representation of the coculture system. 293 T GFP-11 donor cells were transfected with the indicated variant spike expression plasmids and cocultivated with IGROV-1, Vero E6 or Vero E6 TMP-1 acceptor cells expressing GFP1-10. The area of GFP+ fused cells was measured after 18 h. Image generated on Biorender. **b** Representative images of cell-cell fusion between 293 T donor cells and IGROV-1 acceptor cells. Scale bar, 200 μm. **c** Fusogenicity of the different spikes with IGROV-1 acceptor cells. Each dot represents a single experiment. Data are mean ± s.d. of 4 (XBB.1.16.1, EG.5.1.1, BA.2.86.1) or 6 independent experiments. One-way ANOVA with Kruskal-Wallis test followed by Dunn's test for multiple comparisons to compare Delta with respective variants were conducted. Delta vs. Empty, p-value < 0.0001; Delta vs. BA.1, p-value = 0.0005; Delta vs. BA.4/5 p-value = 0.0266; Delta vs. XBB.1.16.1, p-value = 0.0107; Delta vs. EG.5.1.1, p-value = 0.0164; Delta vs. BA.2.86.1, p-value = 0.0164. **d** Effect of TMPRSS2 on the fusion of the different spikes. Cell-cell fusion assays were performed with Vero E6 or Vero E6 TMP-1 as target cells. Data are mean ± s.d. of 4 independent experiments. Paired t-test to compare fusion in Vero E6 versus Vero E6 TMP-1 were conducted. D614G, p-value = 0.0155; BA.1, p-value = 0.0260; BA.4/5, p-value = 0.0136; BQ.1.1, p-value = 0.0071; XBB.1.5/1.9.1, p-value = 0.001; XBB.1.16.1, p-value = 0.0172; EG.5.1.1, p-value = 0.0208; BA.2.86.1, p-value = 0.0111. **e,f** Binding of soluble ACE2 to IGROV-1 infected cells (**e**) or to 293 T cells transiently expressing the Spike (**f**). Cells were stained with serial dilutions of soluble ACE2. The EC50 of ACE2 binding (μg/ml) for the indicated spike proteins is shown. Data are mean ± s.d. of 3 (**f**) or 4 (**e**) independent experiments. One-way ANOVA with Kruskal-Wallis test followed by Dunn's test for multiple comparisons to compare Delta (**e**) or D614G (**f**) with respective variants were conducted. **e** Delta vs. BA.2.86.1, p-value = 0.0292. **f** D614G vs. BA.4/5, p-value = 0.0438; D614G vs. BA.2.86.1, p-value = 0.0017. Source data are provided as a Source Data file.

We then asked how a boost with the bivalent original/BA.5 Pfizer mRNA vaccine modified these humoral responses. Sera from 20 and 15 individuals were tested 1 month and 6 months (Tables S2b, c) after the bivalent boost, respectively (Fig. 6b, c). After 1 month, titers were increased against all tested variants, when compared to individuals that did not receive the boost (Fig. 6a, b). The highest responses were detected against D614G, and to a lower extent to BA.1 and BA.5, reflecting a probable imprinting of the immune response. The responses against the recent XBB-derived and BA.2.86 variants remained about 10–25 fold lower than against BA.5. A similar trend was observed 6 months after the boost. Neutralization was reduced against all strains, highlighting the declining humoral response[53,54]. The neutralizing activity was barely detectable against XBB.1.16.1, EG.5.1.3 and BA.2.86.1 (Fig. 6c). We confirmed the stimulation and subsequent decline of the response elicited by the boost by longitudinally analyzing 15 individuals, tested before and one, three and 6 months after vaccine administration. The neutralizing titers peaked at 1 month and then progressively declined over time (Fig. 6d). The decreases of neutralization titers for all variants, compared to D614G, in the various categories of sera, are depicted Table S4.

Altogether, these results indicate that the XBB-derived and BA.2.86.1 variants are poorly neutralized by sera from individuals having received 3 doses of the original monovalent vaccine. A bivalent boost increased neutralizing titers, which remained however low with EG.5.1.3, XBB.1.16.1 and BA.2.86.1.

### Impact of XBB breakthrough infections on neutralization
We then examined the impact of breakthrough infections on the cross-neutralizing activity of serum antibodies. We analyzed the sera from twelve individuals that were infected in September 2023 (Table S2d), at a time where the main variants circulating in France were XBB.1.9*, XBB.2.3* and XBB.1.16* (representing 49%, 20%, and 14% of the sequenced cases respectively) (GISAID https://www.epicov.org). Samples were analyzed about 3 weeks post-infection (median 19 days; range 10–50 days). A strong augmentation of neutralization against all viruses tested was observed, with ED50 between $1.5 \times 10^3$–$1.4 \times 10^4$ (Fig. 6e). Compared to BA.1, the Nab titers were reduced by about 4-fold against EG.5.1.3 and BA.2.86.1 (ED50 of $2 \times 10^3$ and $1.4 \times 10^3$ respectively). Therefore, post-vaccination breakthrough infection during circulation of XBB-related viruses led to an increase in neutralizing antibody titers, with reduced disparities between variants. This suggests that the anamnestic humoral response triggered by XBB infection includes both a recall of a B-cell memory and the induction of a cross-neutralizing immunity.

### Comparison of BA.2.86.1 and JN.1 variants
BA.2.86 worldwide circulation has been associated with a rapid diversification. Multiple sublineages, some carrying substitutions in the spike, have been designated (Fig. 7a). As mentioned above, the JN.1 sublineage became predominant worldwide in December 2023-Januray 2024. We thus compared BA.2.86.1 and JN.1 fusogenicity, affinity to ACE2, replication in hNECs and sensitivity to immune sera (Fig. 7). The fusogenic potential of BA.2.86.1 and JN.1 was globally similar, as assessed by visualizing the syncytia formed upon infection of S-Fuse cells (Fig. 7b). The affinity to ACE2 was measured as described above. The EC50 of ACE2 binding was enhanced by 1.8-fold for JN.1, indicating a decreased affinity for the receptor (Fig. 7c). In hNECs, both variants replicated with no significant differences observed at 24 h p.i. (Fig. 7d). The sensitivity of JN.1 to neutralization by sera from individuals having received three doses of the original Pfizer vaccine was particularly low, with about a 2-fold decrease in EC50 compared to BA.2.86.1 (Fig. 7e and Table S4). A similar trend was observed after a BA.5 bivalent vaccine boost, which triggered a moderate and short-lasting increase in NAb levels (Fig. 7e). However, a breakthrough XBB infection enhanced neutralization titers to ED50 about $10^3$, with no significant differences between BA.2.86.1 and JN.1 (Fig. 7e). Therefore, as described above with other recently circulating XBB-derived variants, an XBB breakthrough infection triggered a cross-protective response allowing a moderate but significant neutralization of JN.1 (Fig. 7e and Table S4).

Thus, JN.1 displays lower affinity to ACE2 and higher immune evasion properties compared to BA.2.86.1, which likely contributes to its success, possibly in combination with epidemiological factors.

## Discussion
We show that the predominant SARS-CoV-2 Omicron variants circulating in the fall 2023 have progressed towards both increased fitness, as visible in primary cell cultures, and enhanced immune evasion properties. Independently, a novel variant has emerged, similarly presenting high fitness and immune evasion despite a distinct constellation of changes. The tropism of the recent variants for cell lines has also changed. Vero E6 cells no longer allowed efficient replication of EG.5.1.1, EG.5.1.3,BA.2.86.1 and JN.1 strains. Addition of TMPRSS2 did not necessarily increase replication of these recent strains in Vero E6 cells, as illustrated by the discrepant results obtained with Vero E6 TMP-1 and TMP-2 clones. Clonal specificities may explain these differences. However, efficient viral replication occurred in Vero E6 TMP-2 cells, that express low levels of TMPRSS2, and not in Vero E6 TMP-1, that express high levels of TMPRSS2. This protease primes the spike for fusion, but also cleaves ACE2[36]. Initial SARS-CoV-2 variants used both cleaved and uncleaved ACE2 as a receptor, whereas a shift towards preferential use of uncleaved ACE2 for Omicron variants has been proposed[55]. Our results strongly suggest that a delicate balance between ACE2 and TMPRSS2 levels is necessary for optimal replication of recent variants in Vero E6 cells.

We previously reported that IGROV-1 cells are highly permissive to previous SARS-CoV-2 strains[35]. We extend here this observation to XBB-derived and BA.2.86.1/JN.1 strains and explore the underlying mechanisms. Another study confirmed the sensitivity of IGROV-1 cells to SARS-CoV-2 and reported that the cells are not defective in their

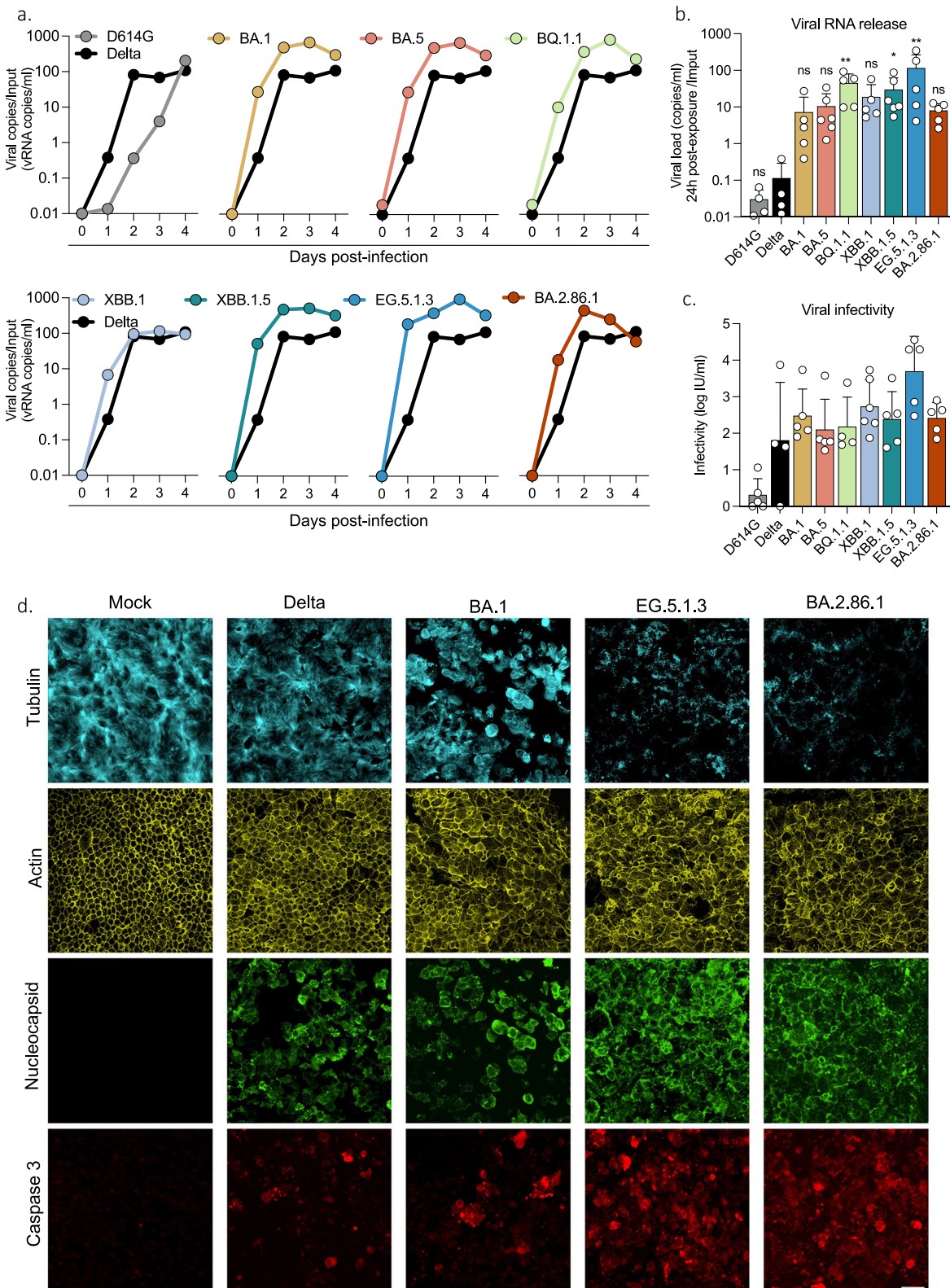

interferon response to infection[56]. We show that IGROV-1 cells naturally express ACE2 but are negative for TMPRSS2, using a sensitive anti-TMPRSS2 nanobody[57]. Moreover, the TMPRSS2 inhibitor Camostat did not impair infection in IGROV-1 but was active in Vero E6 TMP-1 cells, confirming the absence of TMPRSS2 in IGROV-1 cells. In contrast, two cathepsin inhibitors, SB412515 and E-64d strongly inhibited viral infection in IGROV-1, but acted poorly in Vero E6 cells and were

inefficient in Vero E6 TMP-1 cells. Our results indicate that the high permissibility of IGROV-1 to all SARS-CoV-2 strains is likely due to an efficient TMPRSS2-independent endocytic viral entry pathway.

We studied the fusogenicity and receptor binding properties of the variant spikes. The EG.5.1, BA.2.86.1 and JN.1 spikes were not more fusogenic than BA.1 and BA.4/5 spikes, and even less than BQ.1.1 and XBB.1.5. The selective advantage of the most recent variants is thus not

**Fig. 4 | Replication of SARS-CoV-2 variants in hNECs.** Primary human nasal epithelial cells (hNECs) cultivated at the air–liquid interface (ALI) were exposed to the indicated SARS-CoV-2 variants. **a** Viral RNA release from the apical side of hNECs was measured by RT-qPCR every day up to 4 days p.i. Replication kinetics of each variant from one representative experiment are represented. **b** Comparison of viral RNA release at day 1 pi with the indicated variants. **c** Infectious viral titers in supernatants from the apical side were quantified with S-Fuse cells at day 2 p.i. b,c. Data are mean ± s.d. of 4 (D614G, Delta), 5 (BA.1, BQ1.1, XBB.1, EG.5.1.3, BA.2.86.1) or 6 (BA.5, XBB.1.5) independent experiments. One-way ANOVA with Kruskal-Wallis

test followed by Dunn's test for multiple comparisons to compare Delta with respective variants were conducted. **c** D614G vs. BQ.1.1, *p*-value = 0.0032; D614G vs. XBB.1, *p*-value = 0.0116; D614G vs. EG.5.1.3, *p*-value = 0.0057.
**d** Immunofluorescence of hNECs stained for tubulin (cyan), actin (yellow), SARS-CoV-2 Nucleocapsid (green) and cleaved caspase-3 (red). Shown is one representative field (150 × 150 mm) of each variant. Images are from one representative experiment out of 2. Scale bar = 20 μm. Source data are provided as a Source Data file.

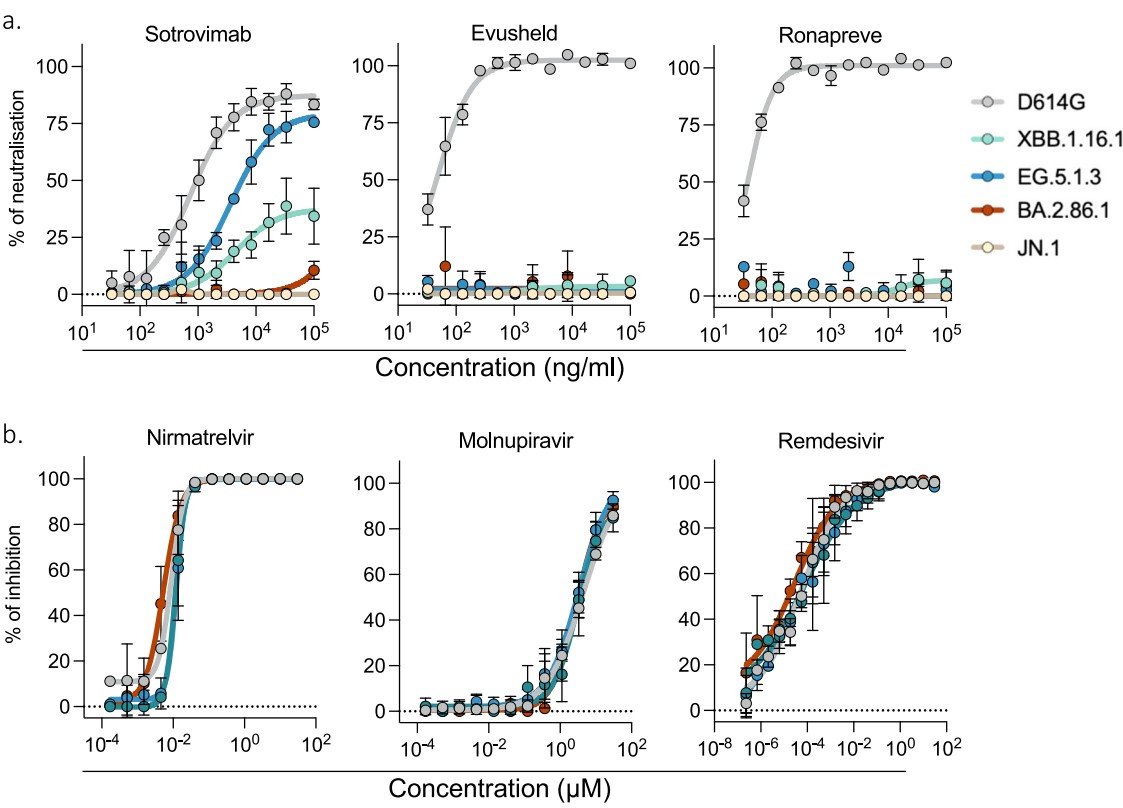

**Fig. 5 | Activity of neutralizing mAbs and antiviral drugs against SARS-CoV-2 variants. a** Neutralization curves of previously approved mAbs. Dose–response analysis of neutralization of the indicated variants by Sotrovimab, Evusheld (Cilgavimab and Tixagevima) and Ronapreve (Casirivimab and Imdevimab).

**b** Inhibitory curves of antiviral drugs against the indicated variants. Dose-response analysis of the antiviral effect of Nirmatrelvir, Remdesivir and Molnupiravir.
**a, b** Data are mean ± s.d. of 3 independent experiments. Source data are provided as a Source Data file.

associated to an increased ability to fuse and to form syncytia, at least in the cell lines tested. Addition of TMPRSS2 in target cells similarly increased fusion with the different spikes, indicating that they remain sensitive to the protease. The BA.2.86.1 spike expressed at the surface of infected cells bound with higher affinity to ACE2 than EG.5.1.3 and XBB.1.16.1 (3.5-fold and 4.3-fold decrease in EC50, respectively). This confirms recent results obtained with recombinant proteins and might be linked to a better exposure of the BA.2.86 RBD than those of XBB-derived or EG.5.1 strains[27]. This increased affinity may contribute to immune escape by itself. In addition, variants with high affinity for ACE2 may have a stronger evolutionary potential as they could tolerate more escape mutations in the RBD despite their negative impact on ACE2 binding, as seen for instance with JN.1.

XBB.1-derived variants, BA.2.86.1 and JN.1 rapidly and potently replicated in primary nasal epithelial cells, amplifying a trend already observed with previous Omicron variants[39,58]. This efficient replication was previously associated with a greater dependency of Omicron on endocytic entry and a lower usage of TMPRSS2 in these cells[39,58], although this last point remains debated[40]. We show here that as soon as 24 h pi, vRNA release of recent variants was up to 380-fold higher than Delta, and 6-fold higher than BA.1. Among the recent variants,

EG.5.1.3 was the fittest and fastest. The cytopathic effect was particularly marked with EG.5.1.3 and BA.2.86.1. Infection with these two variants was associated with a strong disappearance of cilia, a phenomenon described with other strains[46], and with caspase activation. This accelerated replication in respiratory cells likely represents an important factor explaining the selective advantage and improved transmissibility of the variants. Future work will help determining how ACE2 affinity, entry pathways, TMPRSS2 usage, likely involving distinct combinations of changes in XBB and BA.2.86, as well as mutations in other proteins are regulating viral fitness in this relevant in vitro model.

The antiviral drugs Paxlovid, Remdesivir and Molnupiravir remain active against XBB-derived and BA.2.86 variants, indicating that this therapeutic arsenal is so far not impacted. However, there is no approved mAb remaining on the market. Sotrovimab, which retained partial antiviral activity against BQ.1.1 and XBB.1.5[49], poorly acted on EG.5.1.3 and no longer binds or inhibits BA.2.86.1 and JN.1. This novel lineage carries numerous mutations known to allow mAb evasion[27], including the K356T substitution conferring resistance to Sotrovimab[50,51]. Novel mAbs are under pre-clinical and clinical development[11,59,60]. It will be worth scrutinizing their antiviral activity against BA.2.86.1/JN.1 and other emerging subvariants.

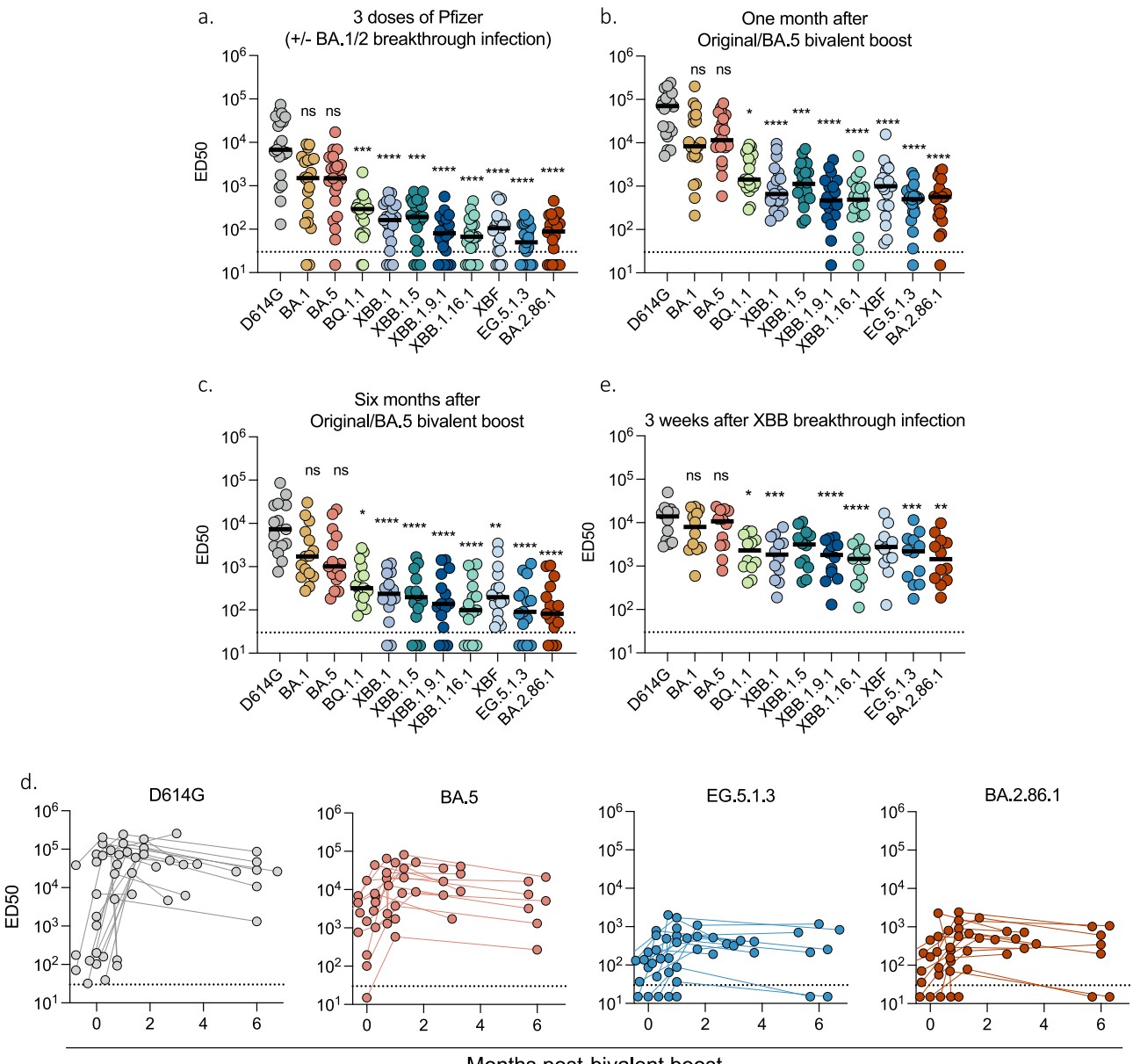

**Fig. 6 | Sensitivity of SARS-CoV-2 variants to sera of vaccinated and/or infected individuals.** Neutralization titers of the sera against the indicated viral variants are expressed as ED50. **a** Neutralizing activity of sera from individuals vaccinated with 3 doses of Pfizer vaccine. Sera ($n = 21$) were sampled 25–475 days after the third dose. 13/21 had a breakthrough infection at the time of BA.1/2 circulation. **b**, **c**. Neutralizing activity of sera from individuals having received the bivalent Original/BA.5 Pfizer boost. Sera were sampled 1 month (b; $n = 20$) and 6 months (c; $n = 14$) after the booster dose. **d** Temporal evolution of Neutralizing Antibody (Nab) titers against D614G, BA.5, EG.5.1.3 and BA.2.86.1 after bivalent Wuhan/BA.5 booster dose. The Nab titers were calculated at the time of injection (month 0) and at the indicated months after injection. **e** Neutralizing activity of sera from Pfizer-vaccinated recipients after XBB-derived breakthrough infections (infections occurred in September 2023, when XBB-derived variants were predominantly circulating in France). Sera were sampled 10–50 days after the breakthrough ($n = 12$). The dotted line indicates the limit of detection (ED50 = 30). Each dot represents the mean of $n = 2$ independent experiments. Black lines represent the median values. Two-sided Friedman test with Dunn's test for multiple comparisons was performed to compare each viral strain to D614G. 3 doses of Pfizer vaccine: D614G vs. BQ.1.1, $p$-value = 0.0008; D614G vs. XBB.1, $p$-value < 0.0001; D614G vs. XBB.1.5, $p$-value < 0.0001; D614G vs. XBB.1.9.1, $p$-value < 0.0001; D614G vs. XBB.1.16.1, $p$-value < 0.0001; D614G vs. XBF, $p$-value < 0.0001; D614G vs. EG.5.1.3, $p$-value < 0.0001; D614G vs. BA.2.86.1, $p$-value < 0.0001. One month after booster dose: D614G vs. BQ.1.1. $p$-value = 0.0165; D614G vs. XBB.1, $p$-value < 0.0001; D614G vs. XBB.1.5. $p$-value = 0.0003; D614G vs. XBB.1.9.1, $p$-value < 0.0001; D614G vs. XBB.1.16.1. $p$-value < 0.0001; D614G vs. XBF. $p$-value < 0.0001; D614G vs. EG.5.1.3. $p$-value < 0.0001; D614G vs. BA.2.86.1. $p$-value < 0.0001. Six months after booster dose: D614G vs. BQ.1.1. $p$-value = 0.0420; D614G vs. XBB.1, $p$-value < 0.0001; D614G vs. XBB.1.5. $p$-value < 0.0001; D614G vs. XBB.1.9.1. $p$-value < 0.0001; D614G vs. XBB.1.16.1. $p$-value < 0.0001; D614G vs. XBF. $p$-value = 0.0018; D614G vs. EG.5.1.3. $p$-value < 0.0001; D614G vs. BA.2.86.1. $p$-value < 0.0001. After XBB-derived breakthrough infections: D614G vs. XBB.1. $p$-value = 0.0017; D614G vs. XBB.1.9.1. $p$-value < 0.0001; D614G vs. XBB.1.16.1. $p$-value < 0.0001; D614G vs. EG.5.1.3. $p$-value = 0.0002; D614G vs. BA.2.86.1. $p$-value = 0.0014. Source data are provided as a Source Data file.

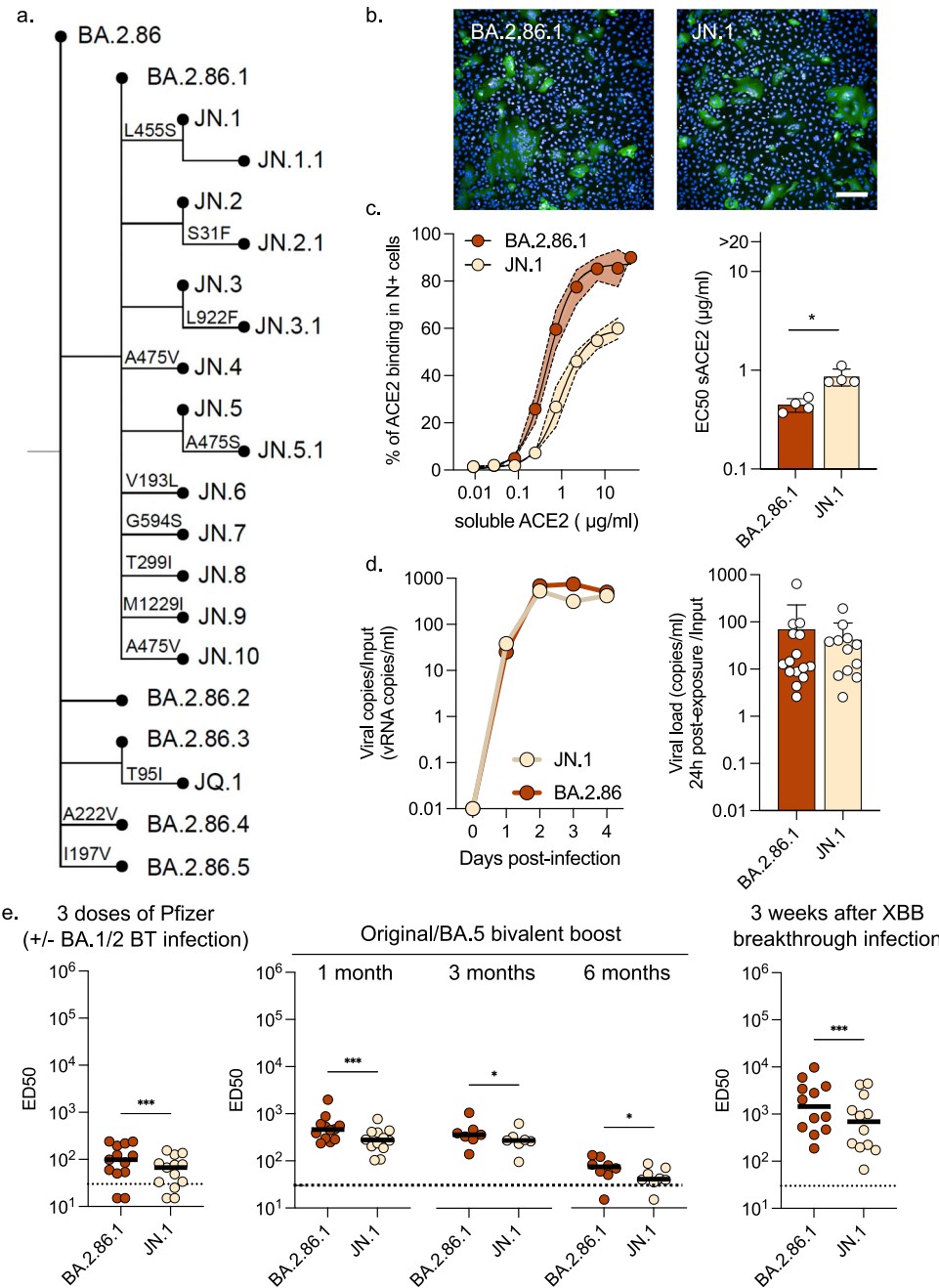

**Fig. 7 | Comparative analysis of BA.2.86.1 and JN.1 variants. a** Schematic tree describing BA.2.86 expanding diversity. Only substitutions in the spike are noted on branches. **b** Representative images of cell-cell fusion in S-Fuse cells after infection with BA.2.86.1 and JN.1. Scale bar, 200 μm. **c** Binding of soluble ACE2 to IGROV-1 infected cells. Data are mean ± s.d (bands) of 2 independent experiments. Infected cells were stained with serial dilutions of soluble hACE2 (left panel). The EC50 of ACE2 (in μg/ml) is displayed (right panel). Data are mean ± s.d of two independent experiments in duplicate. A Two-side Mann-Whitney test was performed. **d** Viral RNA release from the apical side of hNECs was measured by RT-qPCR every day up to 4 days p.i. Replication kinetics of BA.86.1 and JN.1 variants from one representative experiment out of 2 are represented (left). Comparison of viral RNA release at day 1 pi with BA.2.86.1 (*n* = 15) and JN.1 (*n* = 12) (right). Data are mean ± s.d. of samples from 4 independent experiments for BA2.86.1 and 2 independent experiments for JN.1. **e** Comparison of neutralization titers against BA.2.86.1 and JN.1 in sera from individuals in the Orléans cohort. Neutralizing activity of sera (*n* = 13) from individuals vaccinated with 3 doses of Pfizer original

vaccine, sampled 25–475 days after the third dose (left panel). 9/13 had a breakthrough infection at the time of BA.1/2 circulation. Neutralization activity of sera from recipients of a bivalent Wuhan/BA.5 booster dose. Sera were sampled at 1 (*n* = 12), 3 (*n* = 7) and 6 (*n* = 8) months after the booster dose (middle panel). Neutralization activity of sera from Pfizer-vaccinated individuals with a breakthrough infection in September 2023, when XBB-derived variants were predominantly circulating in France (*n* = 12) (right panel). Sera were sampled 10–50 days post-breakthrough infection. The dotted line indicates the limit of detection (ED50 = 30). Each dot represents the mean of *n* = 2 independent experiments. Black lines represent the median values. c,d,e: A Two-side Mann-Whitney test was performed. **c** BA.2.86.1 vs. JN.1, *p*-value = 0.0286; **e.** 3 doses of Pfizer: BA.2.86.1 vs. JN.1, *p*-value = 0.0010; 1 month after booster dose: BA.2.86.1 vs. JN.1, *p*-value = 0.0005; 3 month after booster dose: BA.2.86.1 vs. JN.1, *p*-value = 0.0156; 6 onth after booster dose: BA.2.86.1 vs. JN.1, *p*-value = 0.0156; After XBB-derived breakthrough infections: BA.2.86.1 vs. JN.1, *p*-value = 0.0005. Source data are provided as a Source Data file.

Sera from individuals who had received three doses of COVID-19 Pfizer BNT162b2 vaccine displayed almost no neutralization activity against the recent XBB-derived or BA.2.86 variants. Bivalent boosters increased neutralization titers but only for an abbreviated period of time. Six months after the boost, the efficacy of the sera against EG.5.1.3 and BA.2.86.1/JN.1 was barely detectable. This confirms reports indicating that these two variants are among the most immune evasive viruses described so far, even if some variations between studies have been observed[16,20–29].

We further show that breakthrough infections that occurred in September 2023, during XBB circulation, triggered a broader cross-neutralizing response than bivalent boosters. The differences between variants were reduced, XBB.1, EG.5.1 and BA.2.86.1 neutralization was globally similar in individuals who experienced such XBB break-through infections. Therefore, in addition to imprinted memory, other mechanisms linked to hybrid immunity, such as the generation of responses targeting novel antigens can be efficacious. This observation is in line with reports indicating that XBB.1.5-containing monovalent mRNA vaccines elicit cross-reactive immune responses targeting EG.5.1 and BA.2.86 variants[61,62]. Neutralization titers were however lower against all XBB-derived and BA.2.86 variants than against BA.4/5[61,62]. There is a debate about the interest of annually administrating vaccines based on the circulating variants, especially in healthy individuals[63]. It will be important to analyze the longevity of the humoral response generated by the recent XBB.1.5-based vaccines, and its link with the duration of clinical efficacy against severe forms of the disease.

We also isolated and analyzed the properties of JN.1, a BA.2.86.1 sublineage carrying the L455S spike substitution, that rapidly expanded worldwide and JN.1 displays noticeable differences relative to BA.2.86.1. Its affinity to ACE2 is decreased and its immune evasion properties are higher. We did not detect major differences in the replication of BA.2.86.1 and JN.1 in primary hNECs. Future work will help understand whether the high transmissibility of JN.1 in humans is only due to immune evasion or also associated with other viral properties.

There are limitations to our study. Firstly, we analyzed a limited number of serum samples. However, the marked differences between variants and groups of individuals allowed statistical analysis. Secondly, this work did not include other emerging XBB sublineages showing further evidence of convergent evolution and increasing in frequency. Multiple lineages (e.g. GK.1.1 or JD.1.1 from XBB.1.5; HK.3 or JG.3 from XBB.1.9.2; JF.1 from XBB.1.16; or DV.7.1 from BA.2.75) have independently acquired the F455L substitution in addition to F486P and F456L, a change parallel to the L455S substitution noted in JN.1. Thirdly, we did not explore the role of individual mutations within the spike or in other viral proteins known to have changed in Omicron variants[64]. Fourthly, the method employed here to assess the affinity of various variants for ACE2 is less precise than Bio-Layer Interferometry (BLI), which requires the use of recombinant soluble proteins. However, our approach offers the advantage of assessing the conformation of the Spike protein on the surface of infected cells. Our results are however consistent with recent reports[28,27]. Further research to investigate the mechanisms associated with the modifications of viral properties described here will be relevant in understanding the processes underlying the evolution of SARS-CoV-2.

In summary, we show that SARS-CoV-2 Omicron variants continuously evolve in a context of the mixed immunity of human populations. The selective advantage of EG.5.1 and BA.2.86.1/JN.1 variants combines both convergent increased fitness and replication in respiratory cells, and resistance to the most prevalent antibodies.

## Methods

Our research fulfills all relevant ethical requirements. An informed consent was obtained from all participants. No statistical methods were used to predetermine sample size and the experiments were not randomized. The investigators were not blinded. Sex or gender analysis was not performed due to the limited number of participants.

### Cohorts

**Serum from bivalent Wuhan/BA.5 Pfizer vaccine recipients or from infected individuals (Orléans, France).** The Neutralizing Power of Anti-SARS-CoV-2 Serum Antibodies (PNAS) cohort is an ongoing prospective, monocentric, longitudinal, observational cohort clinical study aiming to describe the kinetics of neutralizing antibodies after SARS-CoV-2 infection or vaccination (ClinicalTrials.gov identifier: NCT05315583). The cohort takes place in Orléans, France and was previously described[65]. This study was approved by the Est II (Besancon) ethical committee. At enrollment, written informed consent was collected, and participants completed a questionnaire that covered sociodemographic characteristics, clinical information and data related to anti-SARS-CoV-2 vaccination.

**Serum from bivalent Wuhan/BA.5 Pfizer vaccinated individuals from Lyon, France.** A prospective, multicentric, longitudinal, interventional cohort clinical study (COVID-SER) is conducted at Hospices Civils de Lyon with the objective to evaluate the effectiveness of commercially developed serological test kits currently in development, which will be used for the diagnosis of patients with suspected SARS-CoV-2 infection. The COVID-SER-VAC ancillary study where blood samples was collected at the time of the injection(s) as per the recommended vaccination schedule was conducted (ClinicalTrials.gov identifier: NCT04341142)[66]. A sub-study aimed to build a collection of biological samples and transfer of residual blood products to external partners for the advancement of scientific knowledge on SARS-CoV-2. We had access to 23 serum samples from hospital staff who were vaccinated with the bivalent Pfizer vaccine. Samples were taken at the time of the injection, 1 month, and 6 months after the injection At enrollment, written informed consent was collected. Virological findings (SARS-CoV-2 RT–qPCR results, date of positive test, screening, or sequences results) and data related to SARS-CoV-2 vaccination (brand product, date of first, second, third and fourth vaccination) were also collected.

**Virus strains.** The D614G, Delta, BA.1, BA.5 and BQ.1.1 and XBB.1.5 strains have been described[35,49,67,68].

The XBB.1 strain (hCoV-19/France/PAC-HCL022171892001/2022) strain was supplied by the National Reference Centre for Respiratory Viruses hosted by the Hôpital de la Croix-Rousse (Lyon, France) and headed by Dr Bruno Lina.

The XBB.1.9.1 strain (hCoV-19/France/GES-IPP08594/2023) was supplied by the National Reference Centre for Respiratory Viruses hosted by Institut Pasteur (Paris, France). The human sample from which strain hCoV-19/France/GES-IPP08594/2023 was isolated, was provided by Dr Djoubi from Massif des Vosges Hospital (France).

The XBB.1.16.1 strain (hCoV-19/France/GES-IPP07712/2023) was supplied by the National Reference Centre for Respiratory Viruses hosted by Institut Pasteur. The human sample was provided by Dr Vanessa Cocquerelle from Laboratory Deux Rives (France).

The XBF strain (hCoV-19/France/IDF-APHP-HEGP-81-10-2332993394/2023) was isolated from a nasopharyngeal swab of individuals attending the emergency room of Hôpital Européen Georges Pompidou (HEGP); Assistance Publique, Hôpitaux de Paris.

The EG.5.1.1 strain (hCoV-19/France/GES-IPP15954/2023) was supplied by the National Reference Centre for Respiratory Viruses hosted by Institut Pasteur. The human sample was provided by Dr Valérie Herzig from Laboratoire Lenys, Colmar (France).

The EG.5.1.3 strain (hCoV-19/France/BRE-IPP15906/2023) was supplied by the National Reference Centre for Respiratory Viruses hosted by Institut Pasteur (Paris, France) The human sample was

provided by Dr F. Kerdavid from Laboratoire Alliance Anabio, Melesse (France).

The BA.2.86.1 strain (hCoV-19/France/IDF-IPP17625/2023) was supplied by the National Reference Centre for Respiratory Viruses hosted by Institut Pasteur. The human sample was provided by Dr Aude Lesenne from Cerballiance, Lisses (France).

The JN.1 strain (hCoV-19/France/HDF-IPP21391/2023) was supplied by the National Reference Centre for Respiratory Viruses hosted by Institut Pasteur. The human sample was provided by Dr Bruno Foucault from Laboratoire Synlab Normandie Maine, La Ferté Macé (France).

All patients provided informed consent for the use of their biological materials. Viral strains were amplified through one or two passages on Vero E6, Vero E6 TMPRSS2, or IGROV-1 cells. Supernatants were harvested 2 or 3 days after viral exposure. The titration of viral stocks was performed on S-Fuse cells[35,41]. Viral supernatants were sequenced directly from nasopharyngeal swabs and after isolation and amplification on Vero E6 or IGROV-1 cells.

For sequencing, an untargeted metagenomic sequencing approach was used, including ribosomal RNA (rRNA) depletion. In brief, RNA was extracted using the QIAamp Viral RNA extraction kit (Qiagen) with the provided poly-A RNA carrier. Prior to library construction, carrier RNA and host rRNA were depleted using oligo(dT) and custom probes, respectively. The resulting RNA from selective depletion was utilized for random-primed cDNA synthesis with SuperScript IV RT (Invitrogen). Second-strand cDNA was generated using Escherichia coli DNA ligase, RNase H, and DNA polymerase (New England Biolabs), and then purified with Agencourt AMPure XP beads (Beckman Coulter). Libraries were prepared using the Nextera XT kit and sequenced on an Illumina NextSeq500 platform (2 × 75 cycles). Reads were assembled using Megahit v1.2.9. The sequences of the stocks of virus isolated in this study have also been deposited on GenBank (Accession numbers: PP405601-PP405606 and PP446819). The complete list of viruses is provided in Table S3.

### Cell lines
IGROV-1, Vero E6 and Vero E6 TMPRSS2 clone 1 (Vero E6 TMP-1) and S-Fuse cells were described previously[41,35]. Vero E6 TMP-2 cells were kindly provided by Dr Makoto Takeda lab[37]. 293 T (CRL-3216) and U2OS (Cat# HTB-96) cells were obtained from ATCC. Cells were authenticated by genotyping (Eurofins). Cells regularly tested negative for mycoplasma.

**Infection of IGROV-1, Vero E6, Vero E6 TMP-1 and Vero E6 TMP-2 cells.** Six hours before infection, 30,000 cells were seeded in a µClear black 96-well plate (Greiner Bio-One). Cells were then infected with the indicated strains of SARS-CoV-2, as described in legend of Fig. 2 and S3 (specifying the variants and quantities of virus used). At days 1 to 4 post-exposure, the cells were fixed using 4% PFA (Electron microscopy cat# 15714-S). The cells were then intracellularly stained with anti-SARS-CoV-2 nucleoprotein (N) antibody NCP-1 (0.1 µg/mL) as described[35]. This staining was carried out in PBS with 0.05% saponin 1% BSA, and 0.05% sodium azide for 1 h. the cells were washed twice with PBS and stained with anti-IgG Alexa Fluor 488 (dilution 1:500, Invitrogen; cat# A11029) for 30 min before being washed twice with PBS. Hoechst 33342 (Invitrogen, cat# H3570) was added during the final PBS wash. Images were captured using an Opera Phenix high-content confocal microscope (PerkinElmer). The N-positive area and the number of nuclei were quantified using Harmony Software v4.9 (PerkinElmer).

**Test of Antiviral molecules.** Cells were seeded in a µClear black 96-well plate (Greiner Bio-One) and pretreated for 2 h with Camostat (Sigma, cat# E8640), E-64d (Sigma, cat# SML0057), SB412515 (Cayman Chemical, cat# 23249), Nirmatrelvir (MedChemExpress; cat# HY-

138687), Remdesivir (MedChemExpress; cat# HY-104077), or Molnupinavir (MedChemExpress; cat# HY-135853) at concentrations as described in the figure legends. Cells were infected with the indicated SARS-CoV-2 strains. After 24 h, infection was revealed as described above. The percentage of inhibition of infection was calculated using the area of N-positive cells as a value with the following formula: $100 \times (1 - (\text{value with drugs} - \text{value in 'non-infected'})/(\text{value in 'no drugs'} - \text{value in 'non-infected'}))$.

The monoclonal antibodies used in this study were previously described[35,68]. Neutralizing activity and ED50 were measured as described in the "S-Fuse neutralization assay" section.

**Plasmids.** SARS-CoV-2 spikes (from D614G, Delta, BA.1, BA.4/5, BQ1.1, XBB.1.5/9, XBB.1.16, EG.5.1, BA.2.86 isolated) were human codon-optimized and produced in silico (GeneArt, Thermo Fisher Scientific) as described[41,43]. Spike sequences were cloned into a phCMV backbone (GenBank: AJ318514) using Gateway cloning (Thermo Fisher Scientific) or restriction enzyme digestion followed by ligation with T4 DNA ligase (New England Biolabs). The pQCXIP-Empty plasmid was used as a negative control[41,43]. All plasmids were sequenced by the Eurofins Genomics TubeSeq service. His-tagged recombinant ACE2 ectodomain (amino acids 19–615) was cloned into pcDNA3.1 vector, produced by transient transfection of HEK293-F cells, and purified by affinity chromatography. Purified ACE2 protein was biotinylated using the EZ-Link Sulfo-NHS-Biotin kit (Thermo Fisher Scientific)[69].

**Donor Acceptor fusion assay.** To assess the fusion of the respective spike constructs, syncytia formation assays were performed[41,43]. Briefly, 293T-GFP-11 cells were transfected in suspension at 37 °C for 30 min. The transfection mix was prepared using Lipofectamine 2000 (Thermo Fisher, Scientific 179) with 50 ng of DNA in a 1:10 ratio of SARS-CoV-2-S and pQCXIP-Empty, respectively, before being added to the cells. Following transfection, cells were washed and resuspended in DMEM with 10% FBS. The level of transfection was quantified by surface staining of Spike with pan-coronavirus mAb10 antibody[35] 18 h post-transfection. Subsequently, 30,000 transfected HEK293T cells were co-cultured with 15,000 IGROV-1-GFP-1-10, VeroE6-GFP1-10, or VeroE6 TMP-1 GFP-1-10 cells per well in a µClear black 96-well plate. 18 h post-transfection, Hoechst 33342 (Invitrogen, cat# H3570) was added to the media at a 1:10,000 dilution and images were acquired using the Opera Phenix High-Content Screening System (PerkinElmer). Analysis was performed using Harmony 191 High-Content Imaging and Analysis Software (PerkinElmer, HH17000012, v.5.0), including the counting of nuclei and the GFP area.

**Soluble ACE2 binding.** IGROV-1 cells were seeded in a 6-well plate 12 h before infection with the indicated SARS-CoV-2 variants. The viral inoculum amount was calculated to achieve 50% of infected cells (N-positive cells) at 24 h post-infection. Afterward, the cells were detached in PBS-EDTA (0.1%) and split into a 96-well plate, with 200,000 cells per well. Cells were incubated with serial dilutions concentrations of a soluble biotinylated human ACE2[69]. Cells were washed twice with PBS and stained with Streptavidin Alexa Fluor 647 (dilution 1:500, Invitrogen; cat# S32357) for 30 min. Cells were then washed twice with PBS and fixed using 4% PFA (Electron microscopy; cat# 15714-S). Cells were then intracellularly stained with anti-SARS-CoV-2 nucleoprotein (N) antibody NCP-1, as described above. Cells were acquired using an Attune NxT Flow Cytometer (Thermo Fisher). Data were analyzed using FlowJo software (BDBioSciences).

**S-Fuse neutralization assay.** U2OS-ACE2 GFP1-10 or GFP 11 cells, also termed S-Fuse cells, become GFP+ when they are productively infected by SARS-CoV-2[41,70]. Cells were mixed (ratio 1:1) and plated at $8 \times 10^3$ per well in a µClear 96-well plate (Greiner Bio-One). The indicated SARS-CoV-2 strains were incubated with serially diluted monoclonal

antibodies or sera for 15 min at room temperature and added to S-Fuse cells. Sera were heat-inactivated for 30 min at 56 °C before use. 18 h later, cells were fixed with 2% PFA (Electron microscopy cat# 15714-S), washed and stained with Hoechst (dilution of 1:1,000, Invitrogen, Invitrogen cat# H3570). Images were acquired using an Opera Phenix high-content confocal microscope (PerkinElmer). The GFP area and the number of nuclei were quantified using the Harmony software (PerkinElmer). The percentage of neutralization was calculated using the number of syncytia as value with the following formula: $100 \times (1 - (\text{value with serum} - \text{value in 'non-infected'})/(\text{value in 'no serum'} - \text{value in 'non-infected'}))$. Neutralizing activity of each serum was expressed as the half maximal effective dilution (ED50). ED50 values (in ng/ml for monoclonal antibodies and in dilution values—i.e titers—for sera) were calculated with a reconstructed curve using the percentage of neutralization at each concentration.

**Human nasal epithelium cells (hNEC) culture, infection and imaging.** MucilAir™, reconstructed human nasal epithelial cells (hNECs) that had been differentiated for 4 weeks prior to obtention, were cultured in 700 µl MucilAirTM media on the basal side of the air/liquid interface (ALI) cultures and monitored for healthy cilia movements. One hour prior to infection, mucus was removed from the apical side of the culture by washing the apical side with warm 200 µl MucilAir™ media. Cells were then infected with equal virus titres in 100 µL MucilAir™ media for 2 h. Viral input was removed and stored at −80 °C. Cells were then washed 2 times for 10 min at 37 °C in warm PBS and then 20 min in 200 µL MucilAir™ media for the day 0 recording. Washing with 100 µl of MucilAir™ warm media was repeated every 24 h for 96 h. Every wash was subsequently centrifuged at 800 g to remove cell debris and frozen at −80 °C. After 96 h, cells were fixed on the apical and basal sides with 4% PFA for 45 min. For imaging, fixed cells were stained intracellularly with anti-SARS-CoV-2 nucleoprotein (N) antibody NCP-1, anti-alpha tubulin (dilution 1:100, 66031-1-Ig; Proteintech), rabbit anti-cleaved caspase-3 (dilution 1:100, D175; Cell Signaling Technology) and phalloidin-Atto 565/633 (dilution 1:500, 75784-1MG-F; Sigma) and imaged using the LSM-700 confocal microscope (Zeiss) as described[57,43].

**Staining of ACE2 and TMPRSS2.** Surface expression of TMPRSS2 and ACE2 was assessed on live cells by staining with anti-TMPRSS2 VHH-A01-Fc[57] at 1 µg/ml or with anti-ACE2 VHH-B07-Fc (Brelot et al., manuscript in preparation) at 0.5 µg/ml, for 30 min at 4 °C in MACS buffer. Then, cells were stained with Alexa Fluor 647-conjugated goat anti-human antibody (Invitrogen; cat# A-21445, 1/500). The control VHH Fc (R3VQFc) recognizes an unrelated protein (phosphorylated Tau protein).

**Phylogenetic tree inference and lineage monitoring.** All available SARS-CoV-2 sequences from human infections were downloaded from the GISAID Epicov database (https://gisaid.org/) on November 27, 2023, and only sequences >29000 nucleotides and with <1% ambiguities (Ns) were kept. SARS-CoV-2 contextual sequence names were retrieved from the Nextstrain build of September 18, 2023[71]. Sequences were reannotated using pangolin (v4.3.1, with option --usher), and aligned against the Wuhan-Hu-1 reference sequence (GenBank MN908947) using nextalign v2.14.0. In addition to this global context, BA.2.86 sequences were added to the dataset, using gofasta v1.2.1[72]. The alignment used for phylogenetic reconstruction was made of 2909 sequences (global context sequences with the addition of BA.2.86 sequences and their closest relatives). Specific positions of the alignment were masked using goalign v0.3.5 (goalign mask command) to decrease phylogenetic noise. Bootstrap alignments were generated using goalign v0.3.5 (goalign build seqboot command) and reference and bootstrap trees were inferred using iqtree v2.2.0 (iqtree -m GTR

-ninit 2 -n 2 -me 0.05 -nt AUTO -ninit 10 -n 4). Bootstrap supports were computed using gotree v0.4.4 (gotree compute support fbp).

Mutations that are common and specific to lineages of interest were computed using the outbreak.info R package (https://outbreak-info.github.io/R-outbreak-info) on January 16, 2023. Values for some insertions or deletions were manually computed. Hierarchical relationships between lineages were retrieved from the pangolin GitHub repository (https://github.com/cov-lineages/pango-designation).

The evolution of the prevalence of SARS-CoV-2 lineages throughout 2023 was visualized using R 4.3 and ggplot 3.4.3, using GISAID data from January 1 to December 31, 2023 (as retrieved on the GISAID EpiCoV database (EPI_SET ID: EPI_SET_231113yq). Workflows used to generate the figures are publicly available at https://github.com/SimonLoriereLab/sarscov2_Oct2023 (https://doi.org/10.5281/zenodo.10692772)).

All genome sequences and associated metadata used to build Fig. 1 and S1 are published in GISAID's EpiCoV database. To view the contributors of each individual sequence with details such as accession number, Virus name, Collection date, Originating Lab and Submitting Lab and the list of Authors, visit https://doi.org/10.55876/gis8.240122bu (Fig. 1a) or https://doi.org/10.55876/gis8.231020kn (Fig. 1b).

**Statistical analysis.** Flow cytometry data were analysed using FlowJo v.10 (TriStar). Calculations were performed using Excel 365 (Microsoft). Figures were generated using Prism 9 (GraphPad Software). Statistical analysis was conducted using GraphPad Prism 9. Statistical significance between different groups was calculated using the tests indicated in each figure legend.

**Reporting summary**
Further information on research design is available in the Nature Portfolio Reporting Summary linked to this article.

## Data availability
All data supporting the findings of this study are available within the article or from the corresponding authors upon reasonable request without any restrictions. The raw data generated in this study are provided in the Source Data file. The sequencing data generated in this study have been deposited on GenBank (Accession numbers: PP405601-PP405606 and PP446819). The bioinformatic workflow used has been deposited on Zenodo (https://zenodo.org/doi/10.5281/zenodo.10692772). Source data are provided with this paper.

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

## Acknowledgements

The authors thank Nicoletta Casartelli and Timothée Bruel for careful reading of the manuscript, the patients who participated to this study, the members of the Virus and Immunity Unit and other teams for dis-cussion and help, Julien Puech from Hôpital Europeen Georges Pom-pidou for organizing the collect and analysis of nasopharyngeal swabs, Hugo Mouquet and Cyril Planchais for the kind gift of reagents, Nathalie Aulner and the staff at the UtechS Photonic BioImaging (UPBI) core facility (Institut Pasteur), a member of the France BioImaging network, for image acquisition and analysis, Fabienne Peira, Vanessa Legros, Aurelie Theillay, Sandra Pallay and Daniela Pires Roteia (CHU Orléans) for their help with the cohorts. The authors thank "the Centre de Ressources biologiques" from Hospices Civils de Lyon (BRIF num-ber: BB-0033-00046), the members of the National Reference Center for viruses of respiratory infections hosted at Institut Pasteur, and the Mutualized platform for Microbiology (P2M). We gratefully acknowl-edge all data contributors, i.e., the Authors and their Originating laboratories responsible for obtaining the specimens, and their Sub-mitting laboratories for generating the genetic sequence and meta-data and sharing via the GISAID Initiative, on which some of this research is based. This work has used the computational and storage services (Maestro cluster) provided by the IT department at Institut Pasteur, Paris. We also acknowledge all the members of the Lyon Covid-ser study group including Jean-Baptiste Fassier, Nicolas Gui-bert, Gregory Destras, Dulce Alfaiate, Amélie Massardier-Pilonchery, Antonin Bal, Hélène Lozano, Bouchra Mokdad, Kahina Saker, Cécile Barnel, Fanny Joubert and Camille Mena and all the members of the clinical research department (DRS) of Hospices Civils de Lyon for their contribution. Work in OS lab is funded by Institut Pasteur, Urgence COVID-19 Fundraising Campaign of Institut Pasteur, Fondation pour la Recherche Médicale (FRM), ANRS, the Vaccine Research Institute (ANR-10-LABX-77), Labex IBEID (ANR-10-LABX-62-IBEID), ANR / FRM Flash Covid PROTEO-SARS-CoV–2, ANR Coronamito, HERA european funding, Sanofi and IDISCOVR. DP is supported by the Vaccine Research Institute. The E.S.-L. laboratory is funded by Institut Pasteur, the INCEPTION program (Investissements d'Avenir grant ANR-16-CONV-0005), the Ixcore foundation for research, the French Govern-ment's Investissement d'Avenir programme, Laboratoire d'Excellence 'Integrative Biology of Emerging Infectious Diseases' (grant no. ANR-10-LABX-62-IBEID), the HERA Project DURABLE (grant no 101102733) and the NIH PICREID (grant no U01AI151758). The Opera system was co-funded by Institut Pasteur and the Région ile de France (DIM1Health). Work in UPBI is funded by grant ANR-10-INSB-04-01 and Région Ile-de-France program DIM1-Health. The funders of this study had no role in study design, data collection, analysis and interpretation, or writing of the article.

## Author contributions

Experimental strategy design, experiments: D.P., I.S., V.M., F.P., F.G-B, W.B., M.H., J.B., T.V., E.S.-L., O.S. Vital materials: F.D., B.J., A.B., O.D., L.A., P.R., D.V., H.P., B.L., S.T-A. L.H., T.P., E.S.-L. Phylogenetic analysis: F.L., E.S.-L. Viral sequencing: M.P., D.V., H.P., B.L., E.S.-L. Manuscript writing and editing: D.P., E.S-L., O.S.

## Competing interests

The authors declare no competing interests.

## Additional information

[1]Virus and Immunity Unit, Institut Pasteur, Université Paris Cité, CNRS UMR3569 Paris, France. [2]Vaccine Research Institute, Créteil, France. [3]Pathogenesis of Vascular Infections Unit, Institut Pasteur, INSERM, Paris, France. [4]G5 Evolutionary Genomics of RNA Viruses, Institut Pasteur, Université Paris Cité, Paris, France. [5]Bioinformatics and Biostatistics Hub, Paris, France. [6]National Reference Center for Respiratory Viruses, Institut Pasteur, Paris, France. [7]Humoral Immunology Laboratory, Institut Pasteur, Université Paris Cité, INSERM U1222 Paris, France. [8]Laboratoire de Virologie, Service de Microbiologie, Hôpital Européen Georges Pompidou, Paris, France. [9]Functional Genomics of Solid Tumors (FunGeST), Centre de Recherche des Cordeliers, INSERM, Université de Paris, Sorbonne Université, Paris, France. [10]Laboratoire de Virologie, Institut des Agents Infectieux, Centre National de Référence des virus des infections respiratoires, Hospices Civils de Lyon, Lyon, France. [11]CIRI, Centre International de Recherche en Infectiologie, Team VirPath, Univ Lyon, Inserm, U1111, Université Claude Bernard Lyon 1, CNRS, Lyon, France. [12]CHU d'Orléans, Service de Maladies Infectieuses, Orléans, France. [13]These authors contributed equally: Delphine Planas, Etienne Simon-Loriere, Olivier Schwartz.
✉e-mail: delphine.planas@pasteur.fr; etienne.simon-loriere@pasteur.fr; olivier.schwartz@pasteur.fr

