## [Peer Review File · Nature Communications]

Distinct evolution of SARS-CoV-2 Omicron XBB and BA.2.86/
JN.1 lineages combining increased fitness and antibody
evasionReviewers' Comments:

Reviewer #1:

Remarks to the Author:

In this study, Planas and his colleagues isolated and cultured XBB.1.5, XBB.1.9.1, XBB.1.16.1, XBF, EG.5.1.1, EG.5.1.3, and BA.2.86.1 live virus, and examined the ability of XBB and BA.2.86 sublineages proliferating in E6 and IGROV-1 cells. They also showed the immune escape of XBB and BA.2.86 sublineages against antiviral antibodies, small molecules and vaccinee and breakthrough infection serum samples. This paper didn't focus on the changes of ACE2 affinity and immune escape brought by certain mutations in spike, and interpretations on some results are questionable.

Specific points

1. In this paper, the authors used different live viruses and recombinant proteins for different experiments, and some of the spike proteins of different variants are the same despite of other parts of the virus (for example XBB.1.9 and XBB.1.9.1, BA.2.86 and BA.2.86.1, etc). Could the authors tie up the names and make them easier to understand and analyse, and also add used variants into Figure 1 and Figure S1 (for example JN.1)?
2. If Figure 1 shows all mutations compared to BA.2, R493Q reverse mutation should be added in the figure.
3. JN.1 has an L455S mutation on BA.2.86 background, which is a very important mutation in terms of immune escape, and the infection rate of JN.1 is on the rise. JN.1 should be added into all section of the results instead of separating it from other variants. More results should be generated on this variant.
4. In Figure S1b, the authors didn't state the meaning of the colour of the squares, do they mean frequency of each mutations?
5. In the experiment of figure S3, the paper stated at page 4 'Viral stocks were serially diluted'. Why didn't the assay start from a certain viral titre so it's more accurate to compare through all variants?
6. In the experiment of Figure S4, can E6 TMP-2 be used in this assay so BA.2.86 and JN.1 can be tested?
7. In page 6, 'The spikes of XBB.1, XBB.1.16.1 and EG.5.1.3 had lower affinities to ACE2 than BA.1'. But from the result the differences between these variants are really minor and insignificant. It's difficult to make the conclusion with this result only. The same applies to the statement in page 7 'The XBB-derived variants exhibited a replication advantage compared to BA.1'.
8. For Figure 4d and 6b, can the authors generate a bar figure of fluorescent area with error bars from biological replicates?
9. In page 7, 'Evusheld and Ronapreve combinations were inactive against the recent variants'. If the authors don't show experimental proof for it, results from other publications should be referred to.
10. Figure S7a should be moved into main figures.
11. There are some confusions with cohort 1 (Table S1a). In page 8, '15 out of 21 individuals experienced a paucity-symptomatic breakthrough infection after the third injection', but #7 and #17 are not, and #4 is uncertain. '14 individuals were infected between Dec 2021 and mid-June 2022', also #7 and #17 are not and #4 is uncertain.

12. In page 8, 'Twenty-one individuals were analysed 12 months post third dose (from cohorts 1 and 2, Table S2a)' should be S2a and b.

13. In page 9, 'which remained however low with EG.5.1.3 and BA.2.86.1', XBB.1.16.1 also showed similar low level of neutralisation and it should be mentioned.

14. The method for measuring virus-ACE2 affinity is not consistent enough. The authors should consider verify these results using alternative ways, for example BLI.

Reviewer #2:

Remarks to the Author:

In the present study, the authors characterized omicron variants that circulated in November 2023, including JN.1. They cultured and sequenced the isolates, noting several mutations in the Spike protein. They then assess replicative ability and fusogenicity in vitro as well as sensitivity against sera from vaccinated and/or infected individuals. Although this is a descriptive study, it is important from a public health standpoint and for future vaccine development to understanding the phenotypes of the dominating variants as well as the robustness of current immunity elicited by vaccination and breakthrough infections. The design of the experiments is sound and the results clearly support the authors' conclusions. However, some additional analysis would strengthen the manuscript.

Comments:

1. Figures 4–6: It would be appropriate to report the geometric mean titer (GMT) values for the neutralization titers. Also, please add the fold-change between D614G and each variant.
2. Figure 5 and Table S1: The current version of Table S1 makes it slightly difficult to follow which samples were used in each figure.
3. It would be informative to compare the replication of BA.2.86.1 and JN.1 in hNECs.
4. Figure S8: Please quantify the flow cytometry data.
5. Please provide a list of the mutations in the viral proteins other than the Spike protein, as they may also impact viral replication.

Reviewer #3:

Remarks to the Author:

REVIEWER COMMENTS

Reviewer #1 (Remarks to the Author):

In this study, Planas and his colleagues isolated and cultured XBB.1.5, XBB.1.9.1, XBB.1.16.1, XBF, EG.5.1.1, EG.5.1.3, and BA.2.86.1 live virus, and examined the ability of XBB and BA.2.86 sublineages proliferating in E6 and IGROV-1 cells. They also showed the immune escape of XBB and BA.2.86 sublineages against antiviral antibodies, small molecules and vaccinee and breakthrough infection serum samples. This paper didn't focus on the changes of ACE2 affinity and immune escape brought by certain mutations in spike, and interpretations on some results are questionable.

We thank Reviewer #1 for appreciating our work.

Specific points

1. In this paper, the authors used different live viruses and recombinant proteins for different experiments, and some of the spike proteins of different variants are the same despite of other parts of the virus (for example XBB.1.9 and XBB.1.9.1, BA.2.86 and BA.2.86.1, etc). Could the authors tidy up the names and make them easier to understand and analyse, and also add used variants into Figure 1 and Figure S1 (for example JN.1)?

We agree that our nomenclature was somewhat confusing. We initially included in the sequence comparison the name of the viral lineage and then used the name of our viral isolate within this lineage. To simplify, we are now using the same names in the different figures. As noted by the reviewer, XBB.1.9 and XBB.1.9.1, BA.2.86 and BA.2.86.1, EG.5.1.1 and EG5.1.3 bear the same spikes, respectively. This is more clearly stated in the text, lines 196-210.

2. If Figure 1 shows all mutations compared to BA.2, R493Q reverse mutation should be added in the figure.

We have corrected our mistake.

3. JN.1 has an L455S mutation on BA.2.86 background, which is a very important mutation in terms of immune escape, and the infection rate of JN.1 is on the rise. JN.1 should be added into all section of the results instead of separating it from other variants. More results should be generated on this variant.

We have now performed novel experiments with JN.1. We have added the description of the JN.1 spike mutations in Fig. 1. We have compared the replication kinetics of JN.1, BA2.86.1 in primary epithelial cells and in IGROV-1, Vero E6 and Vero E6 TMP-2 cell lines. We have tested the sensitivity of JN.1 to the set of monoclonal antibodies that we previously tested on other variants. We have included some of the experiments with JN.1 in some of the figures (Fig. 5, 7, S4, S6). We however prefer to maintain a figure in which we compare JN.1 to its direct predecessor (BA.2.86.1). Our aim is to facilitate the reading of the manuscript. This also allows to better understand why JN.1 supplanted the previous variants, and particularly BA.2.86.1.

We have also included the term JN.1 in the title of the manuscript, that now reads:

“Distinct evolution of SARS-CoV-2 Omicron XBB and BA.2.86/JN.1 lineages combining increased fitness and antibody evasion”

4. In Figure S1b, the authors didn't state the meaning of the colour of the squares, do they mean frequency of each mutations?

Yes. This information has now been added in the legend of Fig. S1, which now reads:
“The color scale reflects the frequency of the mutations within lineages based on the data available on the GISAID EpiCoV database.”

5. In the experiment of figure S3, the paper stated at page 4 'Viral stocks were serially diluted'. Why didn't the assay start from a certain viral titre so it's more accurate to compare through all variants?

We agree with Reviewer #1 that titrating the different viral stocks is an important point. The difficulty is that each variant may display different titers in the various cell lines tested, as shown in this manuscript. To circumvent this issue, we produced all the variants in IGROV-1 cells, and titrated the viral supernatants, over a wide range of dilutions, in both IGROV-1 and Vero E6 cells. The same differences between variants are observed in the two cell lines. When we compared the replication of the different variants in primary cells, we thus used a similar viral input (in terms of virus infectious titers calculated in S-Fuse and IGROV-1 cells). The text in the methods has been modified to describe our titration procedure more clearly. In the experiment presented in Fig. S4, we illustrated the variations in viral replication between IGROV-1 cells and Vero E6 cells. The objective was to highlight the significant differences in viral titers observed depending on the cell line used. We clarified with this sentence line 142: “We characterized the fitness of 9 variants by assessing their replication in different cells and adding as controls D614G, BA.1, BA.5 or BQ.1.1.”

6. In the experiment of Figure S4, can E6 TMP-2 be used in this assay so BA.2.86 and JN.1 can be tested?

We have now compared the infectivity of BA.2.86 and JN.1 in Vero E6 TMP-2 cells, but also in IGROV-1 cells and in primary epithelial cells. The results with Vero E6 TMP-2 and IGROV-1 are presented in Fig. S6. The text now reads line 188:

“We then assessed the sensitivity of Delta, BA.2.86.1 and JN.1 to Camostat (100 μ M), SB412515, and E-64d (10 μ M) in IGROV-1 and Vero E6 TMP-2 cells. SB412515 and E-64d effectively inhibited the replication of the three variants in IGROV-1 cells. Camostat inhibited the replication of the replication of Delta, BA.2.86.1 and JN.1 in Vero E6 TMP-2 cells, but not in IGROV-1 cells, confirming the results observed with the other variants (Fig. S6)”.

7. In page 6, 'The spikes of XBB.1, XBB.1.16.1 and EG.5.1.3 had lower affinities to ACE2 than BA.1'. But from the result the differences between these variants are really minor and insignificant. It's difficult to make the conclusion with this result only. The same applies to the statement in page 7 'The XBB-derived variants exhibited a replication advantage compared to BA.1'.

We agree with Reviewer #1 that the differences between variants are minor, except for BA.2.86.1. As shown in Fig. 3e (infected IGROV-1 cells), BA.2.86.1 displayed a statistically significant lower affinity for ACE2 than D614G, Delta and XBB.1. In Fig. 3f (Spike-transfected 293T cells), BA.2.86.1 displayed a statistically significant lower affinity for ACE2 than D614G and BA.1. We have toned down our message.

The text now reads lines 215-218:

“The spikes of XBB.1, XBB.1.16.1 and EG.5.1.3 had comparable affinities to ACE2 than Delta, whereas BA.2.86.1 bound more potently to the receptor (Fig. 3e). Similar results were observed in 293T cells transiently expressing the different spikes (Fig. 3f), confirming recent reports obtained with recombinant proteins.”

And lines 237-240:

“Therefore, Omicron variants, particularly BQ.1.1 and XBB-derived isolates, exhibit higher infectivity in hNECs compared to D614G and Delta. Among them, EG.5.1.3 demonstrates the highest fitness. Additionally, both EG.5.1.3 and BA.2.86.1 variants exhibit significant cytopathic effects in these cells (Fig. S9).”.

8. For Figure 4d and 6b, can the authors generate a bar figure of fluorescent area with error bars from biological replicates?

We have now added a novel Fig. S9 including a bar representation of the fluorescent areas with error bars from 2 biological replicates (5 random fields for each replicate).

The text now reads lines 239-240:

“Additionally, both EG.5.1.3 and BA.2.86.1 variants exhibit significant cytopathic effects in these cells (Fig. S9).”.

9. In page 7, 'Evusheld and Ronapreve combinations were inactive against the recent variants'. If the authors don't show experimental proof for it, results from other publications should be referred to.

We have now included in Fig. 5 results with Evusheld and Ronapreve combinations, showing that the two cocktails no longer inhibit BA.2.86.1 and JN.1 variants. We also quote recent articles with similar results, although mostly obtained with viral pseudotypes. The text now reads lines 249-254:

“We assessed with the S-Fuse assay the sensitivity of D614G, XBB.1.16.1, EG.5.1.3, BA.2.86.1 and JN.1 to Ronapreve, Evusheld or Sotrovimab. We included the ancestral D614G strain as a control, which was efficiently neutralized by the mAbs (Fig. 5). Evusheld and Ronapreve combinations were inactive against the recent variants. Sotrovimab remained weakly functional against XBB.1.16.1 and EG.5.1.3 but lost antiviral activity against BA.2.86.1 and JN.1.”

10. Figure S7a should be moved into main figures.

We have moved Fig. S7a into the main text. It now appears as Fig. 5.

11. There are some confusions with cohort 1 (Table S1a). In page 8, '15 out of 21 individuals experienced a paucity-symptomatic breakthrough infection after the third injection', but #7 and #17 are not, and #4 is uncertain. '14 individuals were infected between Dec 2021 and mid-June 2022', also #7 and #17 are not and #4 is uncertain.

We apologize for this confusion. The text has been modified and now reads, lines 274-275:

“13 out of 21 individuals experienced a pauci-symptomatic breakthrough infection after the third injection.

And lines 276-277: “13 individuals were infected between December 2021 and mid-June 2022, a period when BA.1 and BA.2 were successively dominant in France ”.

12. In page 8, 'Twenty-one individuals were analysed 12 months post third dose (from cohorts 1 and 2, Table S2a)' should be S2a and b.

The individuals analyzed 12 months post-third dose are now precisely described in Table S2a.

13. In page 9, 'which remained however low with EG.5.1.3 and BA.2.86.1', XBB.1.16.1 also showed similar low level of neutralisation and it should be mentioned.

We have added the following sentence, lines 315-316.:

“...which remained however low with EG.5.1.3, BA.2.86.1 and XBB.1.16.1”.

14. The method for measuring virus-ACE2 affinity is not consistent enough. The authors should consider verify these results using alternative ways, for example BLI.

Regarding the affinity of ACE2 to the different variants, we used the same method that we reported in our Nat. Med. article published in 2021 (ref 71) to describe previous variants. The advantage of the method is that we don't need to produce recombinant proteins, since we measure the binding of a soluble ACE2 molecule to the surface of infected cells. We now quote three articles, including one published a couple of weeks ago in Cell, that measured ACE2 affinity to the spike of BA.2.86 and previous variants (ref 31). The results are similar to our own results. This limitation of our study is now mentioned in the discussion, lines 445-449:

“Fourthly, the method employed here to assess the affinity of various variants for ACE2 is less precise than Bio-Layer Interferometry (BLI), which requires the use of recombinant soluble proteins. However, our approach offers the advantage of assessing the conformation of the Spike protein on the surface of infected cells. Our results are however consistent with recent reports (29,28).”

Reviewer #2 (Remarks to the Author):

In the present study, the authors characterized omicron variants that circulated in November 2023, including JN.1. They cultured and sequenced the isolates, noting several mutations in the Spike protein. They then assess replicative ability and fusogenicity in vitro as well as sensitivity against sera from vaccinated and/or infected individuals. Although this is a descriptive study, it is important from a public health standpoint and for future vaccine development to understanding the phenotypes of the dominating variants as well as the robustness of current immunity elicited by vaccination and breakthrough infections. The design of the experiments is sound and the results clearly support the authors' conclusions. However, some additional analysis would strengthen the manuscript.

We thank reviewer #2 for positively reviewing our manuscript.

Comments:

1. Figures 4–6: It would be appropriate to report the geometric mean titer (GMT) values for the neutralization titers. Also, please add the fold-change between D614G and each variant.

We agree that the GMT values would also be appropriate to report the neutralization titers. We prefer maintaining the median values, because we did so in our previous articles on other variants. This consistent representation facilitates comparisons between our different articles.

As suggested by Reviewer #2, we have now added the fold-change between D614G and each variant in a novel Table S3. This is now stated in the text line 311.

“ The decreases of neutralization titers for all variants, compared to D614G, in the various categories of sera, are depicted Table S4.”

2. Figure 5 and Table S1: The current version of Table S1 makes it slightly difficult to follow which samples were used in each figure.

We agree that the previous version of the Tables S1 and S2 did not allow to fully understand which samples were used in each figure. We have now added a section in Table S2 that indicates the origin of the samples used in each figure. Samples used in Fig. 6a-d are now described in tables S2a-d, respectively.

3. It would be informative to compare the replication of BA.2.86.1 and JN.1 in hNECs.

We have now compared the replication of BA.2.86.1 and JN.1 in hNECs. JN.1 replicates similarly than BA.2.86.1 and more efficiently than Delta in these cells. The experiments now appear in Fig. 7 and are described in the text lines 342-343:

“In hNECs, both variants replicated with no significant differences observed at 24 hours p.i. (Fig. 7d).”

And in the discussion lines 393-394:

“XBB.1-derived variants, BA.2.86.1 and JN.1 rapidly and potently replicated in primary nasal epithelial cells, amplifying a trend already observed with previous Omicron variants”

4. Figure S8: Please quantify the flow cytometry data.

We quantified the flow cytometry data, that are now represented in Fig. S10 and described lines 257-258: “The levels of binding with the different antibodies were quantified (Fig. S10d).”

5. Please provide a list of the mutations in the viral proteins other than the Spike protein, as they may also impact viral replication.

We now provide a novel Fig. S2, which lists the other mutations in BA.2.86 and JN.1 variants. The text now reads lines 123-124:

“JN.1 carries one additional amino acid substitution (F455S) in the spike (Fig. S1b) along with the ORF1a:R3821K and ORF7b:F19L changes (Fig. S2).”

Reviewer #3 (Remarks to the Author):

We thank Reviewer #3 and hope that she/he enjoyed the training!

Reviewers' Comments:

Reviewer #1:

Remarks to the Author:

In this study, Planas and his colleagues isolated and cultured XBB.1.5, XBB.1.9.1, XBB.1.16.1, XBF, EG.5.1.1, EG.5.1.3, and BA.2.86.1 live virus, and examined the ability of XBB and BA.2.86 sublineages proliferating in E6 and IGROV-1 cells. They also showed the immune escape of XBB and BA.2.86 sublineages against antiviral antibodies, small molecules and vaccinee and breakthrough infection serum samples. After revision, the authors addressed most of my questions.

There are some minor corrections needed:

1. Figure 3c and 3d are at the wrong position.
2. Line 276, I counted 12 individuals were infected between December 2021 and mid-June 2022.
3. Line 293, Fig. 5a should be Fig. 6a.
4. Line 301, Tables S2b, c should be Tables S2a, b.

In general I'm happy with this paper.

Reviewer #2:

Remarks to the Author:

The authors responded to my review appropriately, for the most part; however, two minor concerns remain, as detailed below.

1. Figures S10c & d: please describe which regions are treated as N+ or S+ cells in the plot.
2. Figures 4a & 7d: please add error bars.

Reviewer #3:

Remarks to the Author:

REVIEWERS' COMMENTS

Reviewer #1 (Remarks to the Author):

In this study, Planas and his colleagues isolated and cultured XBB.1.5, XBB.1.9.1, XBB.1.16.1, XBF, EG.5.1.1, EG.5.1.3, and BA.2.86.1 live virus, and examined the ability of XBB and BA.2.86 sublineages proliferating in E6 and IGROV-1 cells. They also showed the immune escape of XBB and BA.2.86 sublineages against antiviral antibodies, small molecules and vaccinee and breakthrough infection serum samples. After revision, the authors addressed most of my questions.

We thank Reviewer #1 for appreciating our work.

There are some minor corrections needed:

1. Figure 3c and 3d are at the wrong position.

We have corrected our mistake.

2. Line 276, I counted 12 individuals were infected between December 2021 and mid-June 2022.

We have corrected our mistake.

3. Line 293, Fig. 5a should be Fig. 6a.

We have corrected our mistake.

4. Line 301, Tables S2b, c should be Tables S2a, b.

We checked, but our annotation is actually correct. Table S2a is described line 290. Table S2b corresponds to sera sampled one month after bivalent boost. Table S2c: 6 months post-bivalent boost.

In general I'm happy with this paper.

Thank you again for taking the time to review twice the manuscript

Reviewer #2 (Remarks to the Author):

The authors responded to my review appropriately, for the most part; however, two minor concerns remain, as detailed below.

We thank Reviewer #2 for appreciating our work.

1. Figures S10c & d: please describe which regions are treated as N+ or S+ cells in the plot.

We have added the gating strategy defining the N+ or S+ regions in one of the plots of Figure S10c,d.

2. Figures 4a & 7d: please add error bars.

In Fig 4a and Fig 7d, we show the kinetics of replication of one representative experiment. We cannot add error bar in this figure. The quantifications and statistics are presented in Fig. 4b and 7d (right panel), respectively.

Reviewer #3 (Remarks to the Author):

Thank you for taking part of this training.